# Med-Scout: Curing MLLMs' Geometric Blindness in Medical Perception via Geometry-Aware RL Post-Training

Anglin Liu [1]  Ruichao Chen [2]  Yi Lu [1]  Hongxia Xu [2]  Jintai Chen[†][1][3]

## Abstract

Despite recent Multimodal Large Language Models (MLLMs)' linguistic prowess in medical diagnosis, we find even state-of-the-art MLLMs suffer from a critical perceptual deficit: **geometric blindness**. This failure to ground outputs in objective geometric constraints leads to plausible yet factually incorrect hallucinations, rooted in training paradigms that prioritize linguistic fluency over geometric fidelity. This paper introduces Med-Scout, a novel framework that "cures" this blindness via Reinforcement Learning (RL) that leverages the intrinsic geometric logic latent within unlabeled medical images. Instead of relying on costly expert annotations, Med-Scout derives verifiable supervision signals through three strategic proxy tasks inspired by the systematic reading and reasoning patterns of clinicians: Hierarchical Scale Localization, Topological Jigsaw Reconstruction, and Anomaly Consistency Detection. To rigorously quantify this deficit, we present Med-Scout-Bench, a new benchmark specifically designed to evaluate geometric perception. Extensive evaluations show that Med-Scout significantly mitigates geometric blindness, outperforming leading proprietary and open-source MLLMs by over **40%** on our benchmark. Furthermore, this enhanced geometric perception generalizes to broader medical understanding, achieving superior results on radiological and comprehensive medical VQA tasks. The project page is: https://github.com/HKUSTGZ-ML4Health-Lab/Med-Scout.

[1]The Hong Kong University of Science and Technology (Guangzhou), Guangzhou, China [2]Zhejiang University, Hangzhou, China [3]The Hong Kong University of Science and Technology, Hong Kong, China. Correspondence to: Jintai Chen <jintaiCHEN@hkust-gz.edu.cn>.

*Proceedings of the 43rd International Conference on Machine Learning*, Seoul, South Korea. PMLR 306, 2026. Copyright 2026 by the author(s).

## 1. Introduction

The deployment of Multimodal Large Language Models (MLLMs) in medical imaging necessitates a fundamental shift in priorities compared to general-domain vision-language tasks (Ye & Tang, 2025). While general-purpose models often prioritize open-ended creativity and linguistic fluency, medical visual perception demands an uncompromising adherence to objective geometric constraints. A clinical AI system must not only interpret high-level semantic features but also rigorously respect the intrinsic geometric constraints of the image, such as relative anatomical scales, precise topology, and pixel-level structural consistency. However, despite the rapid evolution of both general (Liu et al., 2023; Bai et al., 2025b;a) and clinically adapted MLLMs (Chen et al., 2024; Team et al., 2025; Sellergren et al., 2025), a critical capability gap remains: while these models excel at generating semantically rich descriptions, they often suffer from *geometric blindness*, failing to ground their outputs in the strict geometric facts.

This misalignment stems largely from the limitations of prevailing training paradigms. Current approaches, including Supervised Fine-Tuning (SFT) and Reinforcement Learning (RL) (Murphy, 2024), predominantly optimize for semantic alignment, maximizing the likelihood of generating clinically plausible text conditioned on visual encodings. While effective for linguistic fluency, these objectives lack explicit mechanisms to penalize violations of geometric constraints. Consequently, models often generate professionally phrased reports that contradict the actual visual evidence. They essentially master complex medical term while failing to perceive the geometric constraints of the scan accurately. Given the need for such strict geometric verification, RL methods (Schulman et al., 2017), specifically Group Relative Policy Optimization (GRPO) (Shao et al., 2024) presents an ideal optimization framework. Unlike standard supervision that relies on likelihood maximization, GRPO enables the model to learn directly from objective feedback signals. This mechanism is particularly effective for instilling geometric awareness, as it allows us to define and enforce explicit constraints that the model must satisfy.

Building upon this perspective, we introduce **Med-Scout**, a geometry-aware post-training framework designed to ac-

tively cure MLLMs' geometric blindness in medical perception. As illustrated in Figure 2, our data-centric approach decomposes medical perception into three geometric proxy tasks: (1) *Hierarchical Scale Localization* formulates the clinical "zoom-in" diagnostic process as a spatial anchoring task, enforcing absolute spatial grounding across varying scales; (2) *Topological Jigsaw Reconstruction* translates anatomical topological deduction into a jigsaw puzzle, demanding a robust understanding of global layouts; and (3) *Anomaly Consistency Detection* frames fine-grained comparative scrutiny as a "spot-the-difference" game, necessitating the precise identification of pixel-level structural discontinuities. To drive the optimization of these tasks, we present a specialized Dense Geometric Reward (DGR) integrated within the GRPO framework, which provides dense guidance, effectively steering the geometric-semantic alignment to ensure stable and balanced convergence across the constructed tasks. To support rigorous training and evaluation, we construct a comprehensive dataset comprising over 100K geometrically perturbed samples and curate a balanced 10% subset to establish Med-Scout-Bench, a novel benchmark that quantifies geometric blindness.

We apply Med-Scout to promote representative general-domain and medical-domain MLLMs. Experiments demonstrate our approach not only significantly reduces geometric blindness but also enables strong generalization to standard medical VQA (Lau et al., 2018; Liu et al., 2021; Zhang et al., 2023c; Hu et al., 2024; Zuo et al., 2025; Butsanets et al., 2025) and report generation tasks (Demner-Fushman et al., 2015; Johnson et al., 2019). This confirms the geometric capabilities acquired effectively transfer to broader medical perception. Our main contributions are as follows:

- *Unveiling Geometric Blindness.* We conduct a pilot study and identify a critical gap where MLLMs fail to ground outputs in geometric constraints despite semantic fluency.

- *Geometry-Aware RL Post-Training.* We propose Med-Scout, a novel RL framework that leverages specialized geometric proxy tasks to actively cure geometric blindness in MLLMs using dense geometric rewards.

- *The Med-Scout-Bench.* We release Med-Scout-Bench, a novel benchmark constructed from the intrinsic geometric properties of medical images. It serves as a rigorous standard for quantifying geometric perception, addressing a key evaluation gap.

- *Substantial Performance Improvements.* Med-Scout improves geometric perception by over **40%** on our benchmark and achieves state-of-the-art generalization on radiological and comprehensive medical VQA benchmarks.

**Conflict of Interest Disclosure.** The authors declare no conflicts of interest.

## 2. Related Work

**MLLMs for the Medical Domain.** The advancement of medical visual understanding is currently driven by two complementary paradigms. On one front, domain-specific adaptation has achieved remarkable precision through biomedical instruction tuning, as demonstrated by a growing array of specialized models (Li et al., 2023; Zhang et al., 2023a; Chen et al., 2024; Team et al., 2025; Pan et al., 2025). Simultaneously, general-purpose foundation models (Liu et al., 2023; Chen et al., 2023; Wang et al., 2025a; Bai et al., 2025a) have exhibited surprising zero-shot adaptability to radiological tasks, benefiting from massive-scale pre-training (Zhang et al., 2023b) and robust visual representations. However, despite these distinct strengths, both paradigms share a critical vulnerability: they frequently prioritize linguistic fluency over physical grounding. This results in pervasive "geometric blindness," where models successfully describe pathologies but fail to adhere to the strict spatial constraints and anatomical layouts inherent in medical images.

**Enhance MLLM with Proxy Tasks.** Recent advancements have integrated verifiable proxy tasks into RL to enhance visual grounding. Jigsaw-R1 (Wang et al., 2025c) and Visual Jigsaw (Wu et al., 2025) established that grid reconstruction significantly improves fine-grained perception, and formalize such objective tasks into RL pipelines. ViCrit (Wang et al., 2025b) employs executable programs for robust verification. Approaches like Euclid (Zhang et al., 2024), GeoPQA (Chen et al., 2025), and GeoGPT4V (Cai et al., 2024) attempt to augment MLLMs with geometric priors, but their task formulations remain incompatible with the unique requirements of medical perception. All these methods fail to address comprehensive clinical constraints like multi-scale consistency and anomaly focus. Moreover, their reliance on sparse binary feedback lacks the granularity essential for complex medical reasoning.

## 3. Pilot Study

We conducted three preliminary experiments on Qwen3-VL-8B-Instruct (Bai et al., 2025a) and Lingshu-7B (Team et al., 2025) to investigate the geometric fidelity of MLLMs, as shown in Figure 1. Our three findings and extensive experiments are described as follows.

### 3.1. Finding I: Inconsistency Between Different Scales

We investigated whether models can consistently integrate visual features across scales. We specifically filtered for 200 positive samples where the models correctly identified lesions in cropped local views. We then evaluated the models on the corresponding original global images of these successfully recognized crops.

**Result:** Models failed to detect the same lesions in over

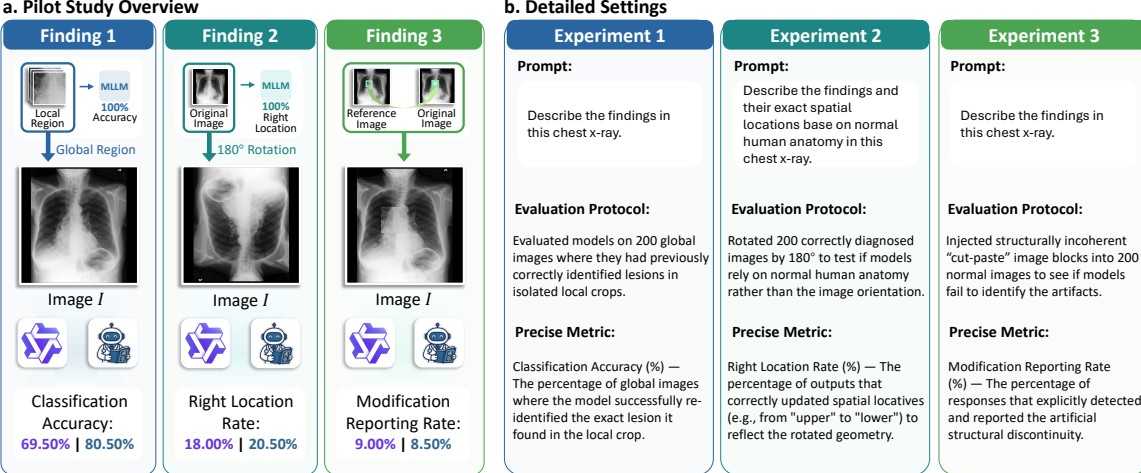

*Figure 1.* Pilot Study: (1) Scale Blindness: The model correctly describes findings in a local crop but fails in the full global view; (2) Topology Blindness: When the image is rotated, the model fails to update location descriptions; and (3) Anomaly Blindness: The model completely overlooks obvious artificial modifications.

20% of the global views. This performance drop indicates the models struggle to preserve visual perception when the spatial scale changes.

### 3.2. Finding II: Blindness to Relative Spatial Positions

We tested whether models verify actual relative spatial positions or rely on semantic priors. We selected 200 images with correctly identified pathology locations and rotated them by 180 degrees. In this inverted view, anatomically "upper" structures appear at the visual bottom.

**Result:** Models failed to adapt their spatial descriptions in 80% of cases. This performance drop indicates the models rely on rigid priors rather than reasoning about the actual topological positions.

### 3.3. Finding III: Insensitivity to Structural Anomalies

We examined sensitivity to pixel-level structural consistency using a "cut-paste" protocol. We artificially replaced image patches with structurally incoherent blocks in 200 samples and prompted models to describe findings.

**Result:** Models failed to identify the artifacts in over 90% of cases. Rather than identifying the artifact, they produced standard reports that entirely overlooked the artificial perturbations. This indicates the models' blindness to abnormal geometric manifestations.

### 3.4. Insufficiency of Chain-of-Thought Reasoning

To determine whether explicit reasoning could alleviate these perceptual deficits, we repeated the three experiments using Chain-of-Thought (CoT) prompting (Table 1). CoT reasoning yielded only marginal performance fluctuations,

*Table 1.* Performance comparison under *No-CoT* and *CoT* modes across the three pilot study experiments.

| MODEL | EXPERIMENT 1 | EXPERIMENT 2 | EXPERIMENT 3 |
|---|---|---|---|
| *No-CoT* | | | |
| QWEN3-VL-8B-INSTRUCT | 69.50% | 18.00% | 9.00% |
| LINGSHU-7B | 80.50% | 20.50% | 8.50% |
| *CoT* | | | |
| QWEN3-VL-8B-INSTRUCT | 67.00% | 20.50% | 12.00% |
| LINGSHU-7B | 77.50% | 22.00% | 7.00% |

failing to fundamentally cure geometric blindness. This indicates that simply enforcing textual reasoning is insufficient without the low-level geometric perception required to ground semantic logic. Crucially, it strongly proves that these failures stem from a fundamental deficit in geometric reasoning capabilities, rather than a mere prompting artifact.

### 3.5. Summary

These experiments reveal critical blindness of current MLLMs: (1) inability to transfer recognition between different scales; (2) blindness to real topological positions; and (3) insensitivity to structural inconsistencies. This semantic-geometric misalignment directly motivates our proposed Med-Scout framework to cure these blind spots, thereby grounding semantic generation in geometric constraints.

## 4. Med-Scout

### 4.1. Task Formulation

We formulate Med-Scout as a geometry-aware RL post-training process, as shown in Figure 2. Our goal is to transform unlabeled medical images into three verifiable geometric proxy tasks $\mathcal{T} = \{\mathcal{T}_{\text{scale}}, \mathcal{T}_{\text{topo}}, \mathcal{T}_{\text{anom}}\}$ as shown

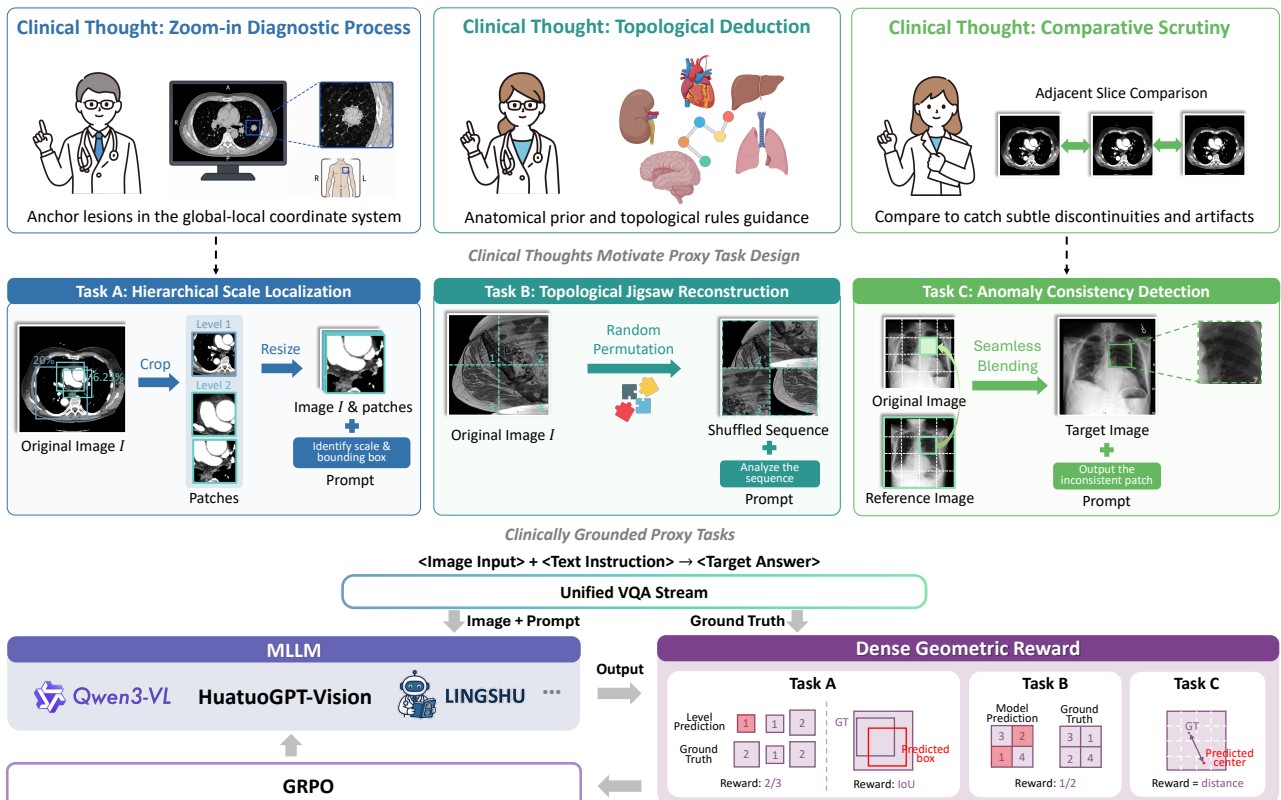

Figure 2. Overview of the Med-Scout Framework. We transform unlabeled medical images into three proxy tasks to cure geometric blindness actively. The framework is optimized using GRPO with a Dense Geometric Reward mechanism that provides stable feedback.

in Appendix A.2. Formally, given a raw medical image $I \in \mathbb{R}^{H \times W}$, we generate a VQA case.

**Task A: Hierarchical Scale Localization ($\mathcal{T}_{\text{scale}}$).**

**Construction:** To enforce multi-scale processing under high cognitive load, we simultaneously crop $N = 3$ local patches from the original image $I$. These patches are sampled from two distinct scale levels: Level 1 (20% of image area) and Level 2 (6.25% of image area). To filter out non-informative background noise, the center coordinates are strictly restricted to the central normalized region (within the $[0.2, 0.8]$ range).

**Objective:** The model identifies the scale level and predicts the normalized bounding box $b = (x_1, y_1, x_2, y_2)$ for each of the $N$ patches. This compels the model to maintain multiple coordinate contexts and master absolute spatial grounding.

**Task B: Topological Jigsaw Reconstruction ($\mathcal{T}_{\text{topo}}$).**

**Construction:** We partition the image $I$ into a $2 \times 2$ grid and apply a random permutation $\sigma$ to generate a shuffled observation $I_{\text{shuffled}}$, retaining sufficient foreground semantic content while requiring both horizontal and vertical spatial reasoning.

**Objective:** The model must reconstruct the sequence of original indices corresponding to the shuffled positions (reading left-to-right, top-to-bottom). This task forces the model to deduce the canonical global layout of anatomical structures through logical geometric deduction.

**Task C: Anomaly Consistency Detection ($\mathcal{T}_{\text{anom}}$).**

**Construction:** We employ a fine-grained "cut-paste" strategy on a $4 \times 4$ grid. We specifically target the core anatomical region by replacing one patch within the central indices with a reference patch from $I_{\text{ref}}$. To ensure realism, $I_{\text{ref}}$ is selected via modality-specific protocols: adjacent slices for volumetric data (CT/ MRI) or the top-1 most similar image retrieved via BiomedCLIP (Zhang et al., 2023b) for X-ray.

**Objective:** The model outputs the grid index of the inconsistent patch. By focusing on the high-density central region with a fine granularity, this task necessitates comparative reasoning to detect subtle pixel-level discontinuities that violate anatomical coherence.

**Unified VQA Stream & Optimization.** We unify these tasks into a standard open-set VQA format as shown in Appendix A.3 and require the model to generate precise answers:

- **Direct Mode:** The model directly generates the final answer.

- **Reasoning Mode:** The model generates a CoT (Wei et al., 2022) reasoning path followed by the answer, allowing it to articulate geometric constraints before concluding.

The final training objective is to maximize the expected reward $\mathcal{R}$ across these generative tasks.

## 4.2. Reward Formulation

To overcome the sparsity of binary feedback in traditional RL, we design a dense geometric reward mechanism. Instead of a simple pass/fail metric, our system calculates continuous reward values based on the degree of geometric deviation, encouraging progressive refinement. The total reward $\mathcal{R}$ is composed of three components:

$$\mathcal{R} = \mathcal{R}_{\text{acc}} + \mathcal{R}_{\text{fmt}} + \mathbb{I}_{\text{CoT}} \cdot \mathcal{R}_{\text{reason}} \quad (1)$$

where $\mathcal{R}_{\text{acc}} \in [0, 1]$ measures task accuracy, $\mathcal{R}_{\text{fmt}} \in [0, 0.5]$ ensures output format compliance, and $\mathcal{R}_{\text{reason}} \in [0, 0.5]$ enforces reasoning structure in CoT mode.

**Dense Geometric Rewards ($\mathcal{R}_{\text{acc}}$).** We tailor specific continuous metrics for each task type to quantify geometric precision, capped at a maximum of 1.0.

- **Scale ($\mathcal{T}_{\text{scale}}$):** This task entails quantifying attributes and localizing targets for $N$ objects simultaneously. We evaluate performance across two dimensions, normalizing by $N$ to ensure invariant optimization magnitude:

  1. *Value Estimation:* For discrete scale levels, we compute the average classification correctness. For a target index sequence $Y^*$ of length $N$:

  $$\mathcal{R}_{\text{val}} = \frac{1}{N} \sum_{i=1}^{N} \mathbb{I}(\hat{y}_i = y_i^*) \quad (2)$$

  2. *Box Localization:* To enforce spatial precision, we calculate the average Intersection over Union (IoU) between the predicted box $\hat{b}_i$ and the ground truth $b_i^*$:

  $$\mathcal{R}_{\text{box}} = \frac{1}{N} \sum_{i=1}^{N} \text{IoU}(\hat{b}_i, b_i^*) \quad (3)$$

- **Topology ($\mathcal{T}_{\text{topo}}$):** To reward partial logical correctness, we utilize an element-wise alignment metric rather than exact sequence matching. For a target index sequence $S^*$ of length $N$:

$$\mathcal{R}_{\text{topo}} = \frac{1}{N} \sum_{i=1}^{N} \mathbb{I}(\hat{s}_i = s_i^*) \quad (4)$$

This ensures the model receives credit for every correctly positioned patch, even if the global sequence is imperfect.

- **Anomaly ($\mathcal{T}_{\text{anom}}$):** The objective is to identify the single swapped patch index $k \in \{0, \ldots, 15\}$ within a $4 \times 4$ grid. To provide dense supervision, we first map the flattened index $k$ to 2D grid coordinates $(u, v)$ via $u = \lfloor k/4 \rfloor$ and $v = k \pmod 4$. We then compute the reward based on the Euclidean distance between the predicted patch coordinates $(\hat{u}, \hat{v})$ and the ground truth $(u^*, v^*)$:

$$\mathcal{R}_{\text{anom}} = \exp\left(-\frac{\sqrt{(\hat{u} - u^*)^2 + (\hat{v} - v^*)^2}}{\tau}\right) \quad (5)$$

where $\tau$ is a temperature hyperparameter. This distance-based mechanism guides the model towards the correct spatial region, rewarding predictions that are geometrically plausible even if the exact index is missed.

**Universal Format Reward ($\mathcal{R}_{\text{fmt}}$).** To strictly enforce output protocols, we evaluate format compliance at the item level rather than the sequence level. Given a target answer containing $N$ required elements, the reward is calculated as:

$$\mathcal{R}_{\text{fmt}} = \frac{0.5}{N} \sum_{i=1}^{N} \mathbb{I}(\hat{a}_i \in \Phi_{\text{regex}}) \quad (6)$$

where $\Phi_{\text{regex}}$ represents the valid pattern for each sub-component.

**Reasoning Structure Reward ($\mathcal{R}_{\text{reason}}$).** Exclusively in Reasoning Mode, we impose an additional structural constraint to enforce the CoT pattern `<think>...<answer>....` This encourages the model to maintain the reasoning buffer:

$$\mathcal{R}_{\text{reason}} = \begin{cases} 0.5 & \text{if } \hat{Y} \in \Phi_{\text{CoT}} \\ 0 & \text{otherwise} \end{cases} \quad (7)$$

Consequently, a perfectly reasoned and accurate response in CoT mode yields a maximum total reward of $\mathcal{R} = 2.0$.

## 4.3. Med-Scout-Bench

To quantitatively evaluate geometric blindness, we introduce Med-Scout-Bench, a novel benchmark encompassing diverse anatomical regions from the mainstream radiological modalities (CT, MRI, and X-ray), pivotal for clinical geometric analysis.

**Dataset Construction.** We synthesize an initial pool of 108,000 VQA cases. We utilize *TotalSegmentor* (CT/MRI) (Wasserthal et al., 2023; Akinci D'Antonoli et al., 2025) to guarantee comprehensive anatomical coverage across the entire body, alongside *MIMIC-CXR* (Johnson et al., 2019) for planar radiography. From this pool, we sampled a high-quality subset of 10,800 cases (10%) as the benchmark. This benchmark is strictly balanced across the three radiological modalities, ensuring unbiased evaluation.

**Evaluation Protocol.** We adopt a unified evaluation setting to ensure consistency: all tasks are formulated as open-set VQA queries rather than multiple-choice options. We employ the LLM-as-a-Judge (Gu et al., 2024) paradigm to robustly assess the semantic correctness of these generative responses, overcoming the limitations of rigid string matching.

## 5. Experiments

### 5.1. Experimental Settings

**Med-Scout Training.** All Med-Scout experiments are conducted for 7,200 steps on a node with $6\times$ NVIDIA RTX PRO 6000 GPUs using GRPO. We optimize the model using AdamW with a peak learning rate of $1 \times 10^{-6}$, a cosine decay schedule, and a warm-up ratio of 0.01. For the GRPO configuration, we set the global batch size to 192 and the group size to $G = 8$, with a KL divergence coefficient $\beta = 0.04$ to ensure stable updates. Training is performed on the 97,200-sample split of our constructed dataset excluding the Med-Scout-Bench. We adopt the LLM-as-a-Judge paradigm using Gemini-3-Flash to evaluate open-ended responses rigorously.

**Baselines.** To assess the generalizability of our framework across different model scales and domains, we apply Med-Scout to four backbones: the general-purpose Qwen3-VL-4B/8B-Instruct (Bai et al., 2025a) and the medical specialists Lingshu-7B (Team et al., 2025) and HuatuoGPT-Vision-7B (Chen et al., 2024). We benchmark our approach against a wide range of state-of-the-art (SOTA) MLLMs, including both proprietary and open-source models.

- **Proprietary Models**: We evaluate the latest commercial leaders, GPT-5 and Gemini-3-Flash, to establish upper-bound performance benchmarks.

- **General-purpose MLLMs**: We select representative open-source models including InternVL3-8B (Zhu et al., 2025), Qwen2.5-VL-3B-Instruct, Qwen2.5-VL-7B-Instruct (Bai et al., 2025b), as well as the newer Qwen3-VL-4B-Instruct and Qwen3-VL-8B-Instruct (Bai et al., 2025a).

- **Medical MLLMs**: For domain-specific comparison, we utilize LLaVA-Med-7B (Li et al., 2023), MedGemma-4B-IT (Sellergren et al., 2025), HuatuoGPT-Vision-7B (Chen et al., 2024), and Lingshu-7B (Team et al., 2025).

To ensure evaluation fairness, all models are assessed within the same standardized evaluation environment.

**Benchmarks.** We conduct comprehensive evaluations across three dataset categories. First, to quantify geometric perception, we utilize our Med-Scout-Bench. Second,

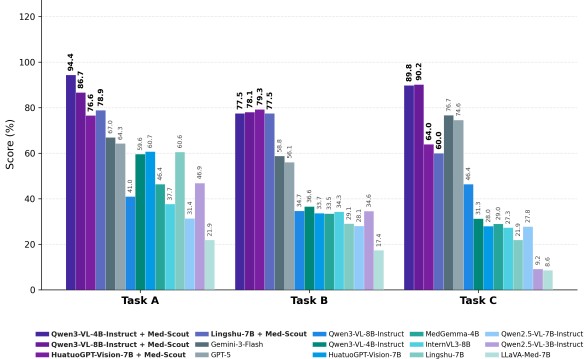

*Figure 3.* Performance comparison on Med-Scout-Bench.

we assess standard radiological VQA using RadImageNet-VQA (Butsanets et al., 2025), VQA-RAD (Lau et al., 2018), and SLAKE (Liu et al., 2021), alongside MIMIC-CXR and IU-Xray for report generation. Finally, to verify broad generalization, we extend our evaluation to PMC-VQA (Zhang et al., 2023c), OmniMedVQA (Hu et al., 2024), and MedXpertQA (Hu et al., 2024) for broader medical domains and diverse modalities.

**Evaluation Metrics.** For Med-Scout-Bench, we report the DGR score computed directly via the reward functions defined in Section 4.2. For VQA benchmarks, we report the response accuracy. For report generation tasks, we report metrics including ROUGE-L (Lin, 2004), CIDEr (Vedantam et al., 2015), and SemScore (Aynetdinov & Akbik, 2024) for both semantic-based and model-based evaluation.

### 5.2. Results on Med-Scout-Bench

Figure 3 illustrates the primary models evaluated in this paper, while Table 6 provides a more detailed performance comparison incorporating additional baseline models on Med-Scout-Bench. As demonstrated in both, Med-Scout yields substantial improvements across all backbones. For instance, the aligned Qwen3-VL-8B-Instruct improves average accuracy from 39.7% to 83.6%, while Lingshu-7B rises from 31.9% to 71.9%. Notably, open-source models outperform leading proprietary models GPT-5 and Gemini-3-Flash. Moreover, general-domain models show greater improvement than medical specialists, suggesting that a strong vision-language foundation enables models to learn geometric features more effectively.

### 5.3. Results on Radiological VQA Benchmarks

Tables 2 and 3 present the evaluation results on radiological VQA and report generation benchmarks, respectively. Med-Scout consistently enhances performance across both discriminative and generative tasks. In VQA, Qwen3-VL-4B achieves a notable 4.2% gain on Rad-VQA, while our aligned Lingshu-7B surpasses Gemini-3-Flash on VQA-

*Table 2.* Performance comparison with SOTA MLLMs across radiological and general medical VQA benchmarks. Rad-VQA represents RadImageNet-VQA. All accuracy metrics are scaled by a factor of 100 to enhance clarity and comprehension. The best results are highlighted.

| MODEL | RADIOLOGICAL VQA | | | GENERALIZATION | | |
|---|---|---|---|---|---|---|
| | RAD-VQA | VQA-RAD | SLAKE | PMC-VQA | OMNIMEDVQA | MEDXPERTQA |
| *Proprietary Models* | | | | | | |
| GPT-5 | 59.1 | 66.4 | 73.9 | 57.7 | 76.9 | 54.8 |
| GEMINI-3-FLASH | 60.7 | 70.2 | 76.1 | **58.1** | 75.3 | **56.0** |
| *General-purpose MLLMs* | | | | | | |
| INTERNVL3-8B | 58.4 | 65.6 | 72.9 | 52.0 | 78.2 | 22.4 |
| QWEN2.5-VL-3B-INSTRUCT | 54.1 | 60.2 | 63.5 | 50.2 | 61.5 | 24.3 |
| QWEN2.5-VL-7B-INSTRUCT | 55.7 | 65.3 | 67.9 | 51.8 | 63.8 | 21.9 |
| QWEN3-VL-4B-INSTRUCT | 41.5 | 59.9 | 73.4 | 42.8 | 45.5 | 27.0 |
| + MED-SCOUT | 45.7↑4.2 | 62.9↑3.0 | 75.6↑2.2 | 45.1↑2.3 | 48.8↑3.3 | 27.7↑0.7 |
| QWEN3-VL-8B-INSTRUCT | 41.6 | 63.2 | 69.6 | 43.9 | 42.9 | 30.4 |
| + MED-SCOUT | 45.3↑3.7 | 65.8↑2.6 | 72.0↑2.4 | 45.5↑1.6 | 46.0↑3.1 | 30.8↑0.4 |
| *Medical MLLMs* | | | | | | |
| LLAVA-MED-7B | 44.3 | 50.6 | 50.1 | 32.4 | 46.8 | 19.9 |
| MEDGEMMA-4B-IT | 49.8 | 70.8 | 77.9 | 48.7 | 70.3 | 22.0 |
| HUATUOGPT-VISION-7B | 48.8 | 67.0 | 67.8 | 53.0 | 75.0 | 22.4 |
| + MED-SCOUT | 52.1↑3.3 | 70.1↑3.1 | 71.0↑3.2 | 55.9↑2.9 | 75.4↑0.4 | 22.7↑0.3 |
| LINGSHU-7B | 61.2 | 68.9 | 82.8 | 56.3 | 81.4 | 27.4 |
| + MED-SCOUT | **64.0**↑2.8 | **71.0**↑2.1 | **83.0**↑0.2 | 57.4↑1.1 | **81.9**↑0.5 | 28.0↑0.6 |

RAD (71.0% vs. 70.2%).

The benefits of geometric alignment are even more pronounced in report generation tasks, where geometric accuracy is critical. As shown in Table 3, Med-Scout boosts Qwen3-VL-4B's CIDEr score by 4.3% on MIMIC-CXR. Most remarkably, Lingshu-7B with Med-Scout achieves an SOTA CIDEr, dramatically outperforming proprietary models. This confirms that curing geometric blindness enables models to generate clinically accurate reports with superior geometric consistency.

### 5.4. Generalization to Comprehensive Medical VQA Benchmarks

We further extend our evaluation to broader medical domains using PMC-VQA, OmniMedVQA, and MedX-pertQA. As shown in Table 2, Med-Scout confers consistent improvements, demonstrating that geometric awareness is fundamental to general medical perception. Notably, HuatuoGPT-Vision-7B achieves a substantial 2.9% gain on PMC-VQA. Even more impressively, Lingshu-7B manages to break through its performance ceiling, further elevating its already SOTA results. This confirms that our framework effectively enhances robustness in complex, open-domain medical perception.

### 5.5. Direct Mode vs. Reasoning Mode

As illustrated in Figure 4, the negligible performance gap between Direct Mode (+M) and Reasoning Mode (+M (R)) suggests that explicit CoT is not strictly necessary. However, this similarity may partially stem from our reasoning structure reward ($\mathcal{R}_{reason}$) prioritizing structural constraints over logical validity.

### 5.6. Data Scaling Analysis

To evaluate the scalability and data efficiency of our framework, we conducted a quantitative analysis across varying training set sizes.

**Med-Scout-Bench Scalability.** As shown in Figure 5 (Left), performance on Med-Scout-Bench improves consistently across all backbone models as the training data volume increases from 20% to 100%. The substantial performance gain in the early stages underscores the high efficiency and quality of our automatically generated supervision signals, while the continuous upward trend without saturation suggests the models have not yet reached their capacity limits.

**Generalization Correlation.** Figure 5 (Right) reveals a strong positive correlation between internal alignment

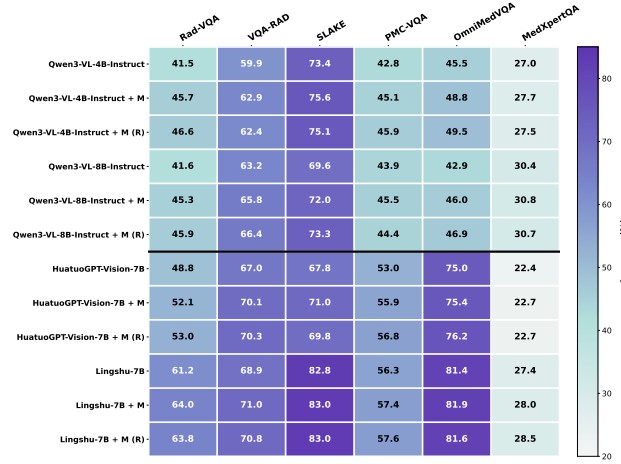

*Figure 4.* Performance comparison on six public benchmarks. Purple colors correspond to higher accuracy. Direct Mode (+M) and Reasoning Mode (+M (R)) show close performance.

*Table 3.* Performance comparison on medical report generation benchmarks. Results are reported on ROUGE-L, CIDEr, and SemScore metrics. All scores are scaled by a factor of 100 to enhance clarity and comprehension. The best results are highlighted.

| MODEL | MIMIC-CXR | | | IU-XRAY | | |
|---|---|---|---|---|---|---|
| | ROUGE-L | CIDER | SEMSCORE | ROUGE-L | CIDER | SEMSCORE |
| *Proprietary Models* | | | | | | |
| GPT-5 | 14.7 | 89.4 | 25.2 | 34.6 | 139.9 | 48.3 |
| GEMINI-3-FLASH | 24.9 | 90.0 | 29.8 | 35.1 | 135.8 | 48.0 |
| *General-purpose MLLMs* | | | | | | |
| INTERNVL3-8B | 20.7 | 64.3 | 21.1 | 22.4 | 70.2 | 30.6 |
| QWEN2.5-VL-3B-INSTRUCT | 21.4 | 57.7 | 19.4 | 26.7 | 72.9 | 37.2 |
| QWEN2.5-VL-7B-INSTRUCT | 22.9 | 63.9 | 18.6 | 26.6 | 78.8 | 36.3 |
| QWEN3-VL-4B-INSTRUCT | 21.8 | 60.9 | 19.8 | 25.8 | 81.4 | 36.5 |
| + MED-SCOUT | 23.4↑1.6 | 65.2↑4.3 | 21.5↑1.7 | 27.1↑1.3 | 84.2↑2.8 | 38.6↑2.1 |
| QWEN3-VL-8B-INSTRUCT | 21.3 | 64.8 | 19.6 | 27.1 | 75.9 | 38.7 |
| + MED-SCOUT | 23.8↑2.5 | 68.1↑3.3 | 21.0↑1.4 | 29.5↑2.4 | 79.6↑3.7 | 41.6↑2.9 |
| *Medical MLLMs* | | | | | | |
| LLAVA-MED-7B | 16.4 | 49.2 | 16.7 | 19.5 | 73.7 | 17.4 |
| MEDGEMMA-4B-IT | 27.4 | 83.8 | 30.1 | 30.6 | 107.5 | 46.8 |
| HUATUOGPT-VISION-7B | 23.6 | 75.6 | 24.6 | 30.9 | 109.6 | 40.7 |
| + MED-SCOUT | 25.7↑2.1 | 79.0↑3.4 | 25.8↑1.2 | 32.1↑1.2 | 111.7↑2.1 | 43.1↑2.4 |
| LINGSHU-7B | 30.9 | 104.9 | 29.7 | 37.7 | 180.8 | 48.4 |
| + MED-SCOUT | 31.4↑0.5 | 105.2↑0.3 | 30.3↑0.6 | 38.0↑0.3 | 183.3↑2.5 | 48.6↑0.2 |

scores and performance on six external benchmarks. This confirms that improving geometric awareness enhances general medical perception. Consequently, Med-Scout-Bench serves as a reliable indicator of broader clinical visual reasoning capabilities.

## 5.7. Comparison with Existing Proxy Tasks

To rigorously evaluate the effectiveness of our proxy task design, we compare Med-Scout against existing visual proxy tasks using both Qwen3-VL-4B-Instruct and Qwen3-VL-8B-Instruct as backbones. While recent methods like Jigsaw-R1 (Wang et al., 2025c) and ViCrit (Wang et al., 2025b) successfully introduce geometric constraints for general-domain models, they fundamentally lack the medical specificity required to handle complex anatomical structures and subtle clinical anomalies. To ensure a strictly fair comparison, we intentionally restrict Med-Scout to use standard sparse rewards, disabling our advanced dense geometric reward (DGR) mechanism. As demonstrated in Table 4, even under this constrained setting, Med-Scout consistently

achieves the highest average accuracy across both radiological VQA and broader medical generalization domains. This compellingly confirms that domain-specific geometric alignment, rather than mere general spatial awareness, is essential for robust medical perception.

*Table 4.* Comparison with existing proxy tasks using Qwen3-VL-Instruct models. We report the average accuracy for Radiological VQA and Generalization benchmarks. Even with sparse rewards, Med-Scout outperforms general-domain tasks.

| METHOD | MED. | GEO. | RAD-VQA (AVG.) | GEN. (AVG.) |
|---|---|---|---|---|
| **QWEN3-VL-4B-INSTRUCT** | | | | |
| BASELINE | - | - | 58.3 | 38.4 |
| + JIGSAW-R1 | ✗ | ✓ | 57.6↓0.7 | 38.3↓0.1 |
| + VICRIT | ✗ | ✗ | 57.7↓0.6 | 38.4↑0.0 |
| + MED-SCOUT | ✓ | ✓ | 60.8↑2.5 | 40.2↑1.8 |
| Δ | - | - | +3.2 | +1.9 |
| **QWEN3-VL-8B-INSTRUCT** | | | | |
| BASELINE | - | - | 58.1 | 39.1 |
| + JIGSAW-R1 | ✗ | ✓ | 57.6↓0.7 | 38.3↓0.1 |
| + VICRIT | ✗ | ✗ | 57.7↓0.6 | 38.4↑0.0 |
| + MED-SCOUT | ✓ | ✓ | 60.4↑2.3 | 40.5↑1.4 |
| Δ | - | - | +3.0 | +1.5 |

## 5.8. Extensive Analysis

We conducted an extensive analysis to evaluate the impact of Med-Scout on the spatial discrimination capabilities of MLLMs and their visual attention focus on target regions.

**Energy Landscape of Factual Consistency.** Following the Energy-Based model (Song & Kingma, 2021), we quantify the compatibility between visual evidence $\mathbf{x}$ and textual description $\mathbf{y}$ via the energy function $E(\mathbf{x}, \mathbf{y}) = -\log P_\theta(\mathbf{y}|\mathbf{x})$, implemented as the negative log-likelihood of the target response. To evaluate this, we constructed a probe dataset of 800 factual-counterfactual pairs from MIMIC-CXR by inverting anomaly-related spatial locatives

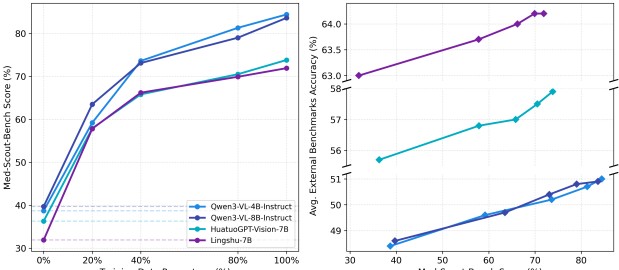

*Figure 5.* Data Scaling and Generalization Analysis. Left: Continuous performance improvement on Med-Scout-Bench with increasing training data. Right: Strong positive correlation between Med-Scout-Bench scores and average accuracy on external benchmarks.

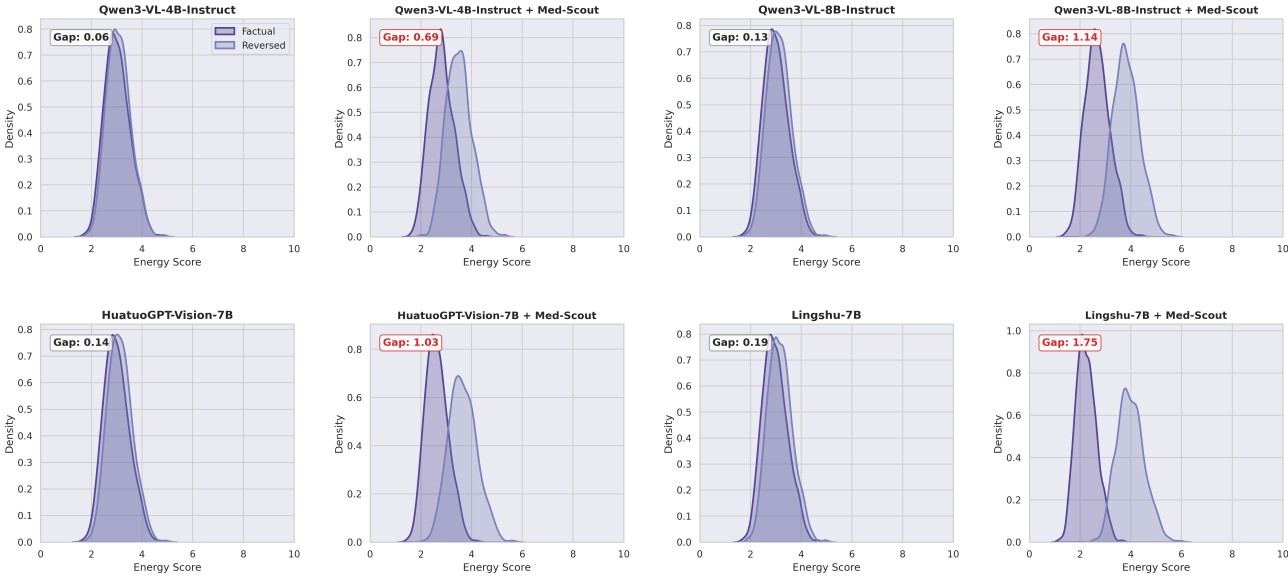

*Figure 6.* Energy Landscape of Factual Consistency. We visualize energy distributions for 800 factual (purple) versus spatially inverted (blue) report pairs. Med-Scout establishes a distinct energy barrier.

in reports. As visualized in Figure 6, the baseline models exhibit a collapsed landscape where factual and perturbed descriptions share overlapping energy distributions, indicating geometric blindness. In contrast, Med-Scout establishes a distinct energy barrier, effectively assigning high-energy states to spatial hallucinations while preserving low energy for factual descriptions. This separation suggests that Med-Scout has successfully internalized the spatial constraints of medical imagery, moving beyond mere language priors to achieve rigorous geometric alignment.

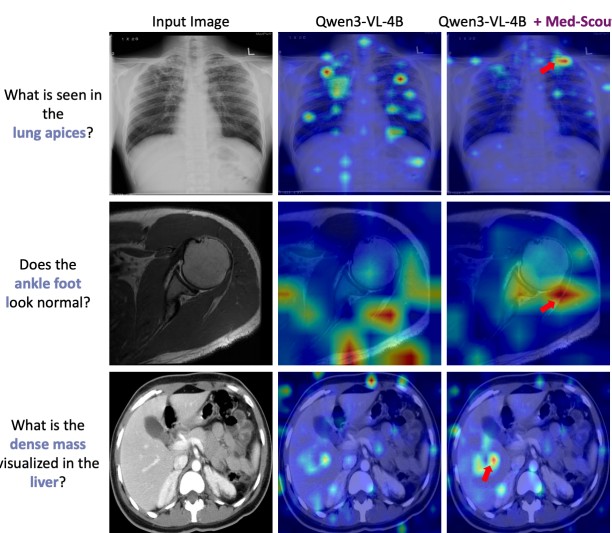

*Figure 7.* Visualization of attention maps on critical anatomical regions. We compare the visual attention of the baseline Qwen3-VL-4B-Instruct (Middle) versus the model aligned with Med-Scout (Right) given specific anatomical queries. Med-Scout demonstrates a highly concentrated focus on the critical target regions.

**Analysis of Attention on Critical Region.** We followed (Zhang et al., 2025) and visualized the attention maps of the MLLM on specific anatomical queries (Figure 7). The baseline model exhibits scattered attention and frequently drifts to irrelevant background noise. In contrast, Med-Scout demonstrates a highly concentrated focus on the critical regions, such as *lung apices*, *ankle joints*, or *dense mass in liver*. This shift confirms that our challenging geometric proxy tasks motivate the model to transition from superficial scanning to fine-grained visual scrutiny, thus effectively enhancing the model's sensitivity to local visual evidence. This improved grounding not only translates directly to the external benchmarks' performance but also provides the reliability essential for medical understanding.

## 6. Conclusion

Motivated by a pilot study revealing the significant "geometric blindness" of MLLMs in medical perception, this paper proposes Med-Scout and Med-Scout-Bench, a geometry-aware RL post-training framework designed to cure this blindness and a novel benchmark aimed to quantify this limitation rigorously. By aligning semantic generation with geometric constraints through three intrinsic proxy tasks and a dense reward mechanism that stabilizes the optimization process, Med-Scout significantly enhances the geometric perception of existing MLLMs and further improves performance on radiological and comprehensive medical VQA benchmarks. Moreover, further analysis reveals the potential of this strategy in maintaining the geometric truth of medical images, thereby substantially enhancing visual capabilities.

## Acknowledgments

This work was supported by the Guangdong Basic and Applied Basic Research Foundation (2026A1515011793), and the Youth S&T Talent Support Programme of Guangdong Provincial Association for Science and Technology (SKXRC2025467), and the Transvascular Implantation Devices Research Institute (KY052025003).

## Impact Statement

This work addresses a fundamental divergence in training priorities. While general MLLMs focus on linguistic fluency, medical applications require strict adherence to geometric constraints. We identify this mismatch as the root cause of "geometric blindness," a critical deficit where models fail to ground their outputs in physical reality.

We resolve this bottleneck with Med-Scout, a cost-efficient post-training framework. By extracting intrinsic geometric rules directly from unlabeled data instead of relying on expensive expert annotations, we use RL to align models with visual logic. Crucially, we prove that curing this specific geometric deficit drives substantial improvements in downstream tasks, such as report generation and medical visual question answering. This establishes a scalable and data-efficient pathway toward reliable clinical AI that strictly respects the physical truth of medical images.

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

# A. Dataset Construction Details

In this appendix section, we provide a detailed breakdown of the data generation pipeline used to construct the dataset for Med-Scout training and the Med-Scout-Bench. Driven by the objective to cure geometric blindness without relying on expensive human annotations, our pipeline automatically extracts verifiable geometric supervision signals directly from raw medical images. We first elaborate on the composition and distribution of the 108,000-sample dataset, which spans diverse modalities including CT, MRI, and X-ray sourced from TotalSegmentor (Wasserthal et al., 2023; Akinci D'Antonoli et al., 2025) and MIMIC-CXR (Johnson et al., 2019). Subsequently, we describe the rigorous algorithmic protocols employed to synthesize the three geometric proxy tasks. Finally, we present the unified VQA instruction templates used to standardize these tasks for effective RL post-training.

## A.1. Data Composition and Distribution Statistics

We primarily analyze the Med-Scout-Bench dataset. This is a specific test set of 10,800 examples designed to accurately measure geometric blindness. The rest of the data used for training and validation ($N = 97,200$) follows the same pattern, ensuring that the way the model is trained matches the way it is tested.

As shown in Figure 8, the benchmark follows two key distribution patterns:

- **Strict Modality Balance:** To avoid bias toward any specific imaging method, we ensure an equal distribution across the three main modalities. The benchmark is split evenly, with CT, MRI, and X-ray each making up about $33.3\%$ of the data.

- **Task-Specific Distribution:** To ensure balanced training, we allocate sample sizes based on the difficulty of each task. We assign 1,800 samples to Task A (Hierarchical Scale Localization), as this fundamental spatial task requires less data to converge. The largest share (5,400 samples) goes to Task B (Topological Jigsaw Reconstruction); its complex anatomical puzzles require robust reasoning, necessitating more data. Finally, we dedicate 3,600 samples to Task C (Anomaly Consistency Detection), an intermediate amount sufficient for the model to learn to detect fine details and subtle anomalies.

## A.2. Task-Specific Generation Protocols

### A.2.1. HIERARCHICAL SCALE LOCALIZATION

This task is designed to compel the model to master absolute spatial grounding and multi-scale consistency. By forcing the model to map resized local patches back to their original global coordinates, we simulate a "zoom-in" clinical diagnostic process where a radiologist examines local details (e.g., a nodule) while maintaining awareness of its global position (e.g., upper right lung lobe). The generation process is formalized in Algorithm 1.

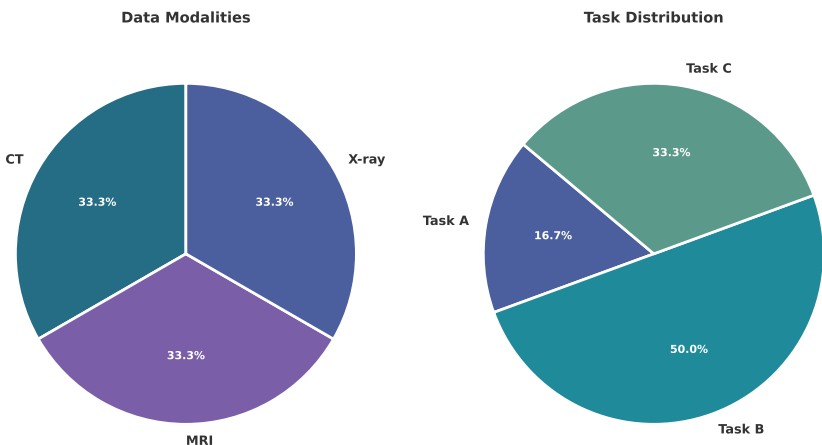

*Figure 8.* Data Statistics of Med-Scout-Bench. Left: The benchmark maintains a strictly balanced distribution across CT, MRI, and X-ray modalities to ensure unbiased geometric evaluation. Right: The benchmark is under task-specific distribution.

**Scale Definitions and Sampling.** We define two distinct scale levels to represent different granularities of anatomical detail:

- **Level 1 (Regional View):** The patch area covers $20\%$ of the original image area ($A_{\text{patch}} \approx 0.2 \times A_{\text{total}}$).

- **Level 2 (Focal View):** The patch area covers $6.25\%$ of the original image area ($A_{\text{patch}} \approx 0.0625 \times A_{\text{total}}$), corresponding to a $4\times$ zoom relative to the original resolution.

For each training instance, we randomly extract $N = 3$ square patches. To ensure the patches contain meaningful anatomical information rather than non-informative background (e.g., black borders common in raw medical scans), the sampling of crop regions is strictly restricted to the valid central interval $[0.2, 0.8]$ of the image dimensions.

**Input and Objectives.** The input to the model consists of the original global image $I_{\text{global}}$ followed by the three resized local patches $\{P_1, P_2, P_3\}$. All patches are resized to the dimensions of $I_{\text{global}}$. The model is prompted to perform two sub-tasks:

1. **Scale Classification:** Identify whether each patch belongs to Scale Level 1 or Level 2.

2. **Coordinate Regression:** Explicitly predict the normalized bounding box $(x_1, y_1, x_2, y_2)$ of the patch in the original image frame.

---

**Algorithm 1** Data Generation for Hierarchical Scale Localization

---

**Require:** Original Image $I \in \mathbb{R}^{H \times W}$
**Require: Hyperparameters:**
  1:     Num Patches $N \leftarrow 3$
  2:     Scale Ratios $\mathcal{S} \leftarrow \{0.20, 0.0625\}$ {Area ratios for Level 1 and Level 2}
  3:     ROI Bounds $[\alpha_{\min}, \alpha_{\max}] \leftarrow [0.2, 0.8]$ {Normalized valid center region}
  4:     Target Size $(H_{\text{out}}, W_{\text{out}}) \leftarrow (H, W)$ {Resize targets to original resolution}
**Ensure:** Resized Patches $\mathcal{P}$, Norm. Coordinates $\mathcal{B}$, Scale Labels $\mathcal{L}$
  5: Initialize $\mathcal{P} \leftarrow [\,], \mathcal{B} \leftarrow [\,], \mathcal{L} \leftarrow [\,]$
  6: **for** $k = 1$ to $N$ **do**
  7:    *// Step 1: Determine Crop Size*
  8:    Sample ratio $r \in \mathcal{S}$ uniformly
  9:    Calculate square side length: $l \leftarrow \sqrt{r \cdot H \cdot W}$
10:    *// Step 2: Calculate Sampling Boundaries*
11:    *// Ensure crop is within ROI: start $\geq \alpha_{min}$, end $\leq \alpha_{max}$*
12:    $x_{\min} \leftarrow \alpha_{\min} \cdot W$
13:    $x_{\max} \leftarrow \alpha_{\max} \cdot W - l$
14:    $y_{\min} \leftarrow \alpha_{\min} \cdot H$
15:    $y_{\max} \leftarrow \alpha_{\max} \cdot H - l$
16:    *// Step 3: Sample and Extract*
17:    Sample top-left $x \sim \text{Uniform}(x_{\min}, x_{\max})$
18:    Sample top-left $y \sim \text{Uniform}(y_{\min}, y_{\max})$
19:    $Patch_{\text{raw}} \leftarrow I[y : y + l, \ x : x + l]$
20:    *// Step 4: Resize and Normalize*
21:    $Patch_{\text{resize}} \leftarrow \text{Resize}(Patch_{\text{raw}}, (H_{\text{out}}, W_{\text{out}}))$
22:    $Box_{\text{norm}} \leftarrow [x/W, y/H, (x + l)/W, (y + l)/H]$
23:    Append $Patch_{\text{resize}}$ to $\mathcal{P}$, $Box_{\text{norm}}$ to $\mathcal{B}$, $r$ to $\mathcal{L}$
24: **end for**
25: **return** $\mathcal{P}, \mathcal{B}, \mathcal{L}$

---

A.2.2. TOPOLOGICAL JIGSAW RECONSTRUCTION

This task tests the model's ability to infer the overall anatomical structure from specific details. Unlike other jigsaw tasks that often use complex, many-piece grids, we use a simple $2 \times 2$ grid. This choice is essential for medical imaging:

- **Semantic Integrity:** A $2 \times 2$ grid ensures that each section contains recognizable anatomical features, such as a complete left lung or the clear shape of the pelvis.

- **Logical Deduction:** It shifts the reasoning burden from low-level pattern matching to high-level topological deduction (e.g., reasoning that the "heart" patch must be spatially adjacent to and above the "stomach" patch).

**Generation Protocol.** The image $I$ is partitioned into four quadrants. We generate a random permutation $\sigma$ of the index set $\{0, 1, 2, 3\}$, where indices correspond to the canonical reading order (top-left, top-right, bottom-left, bottom-right). The quadrants are rearranged according to $\sigma$ to form the shuffled observation $I_{\text{shuffled}}$. The model is tasked with reconstructing the sequence of original indices. The detailed generation process is described in Algorithm 2.

---

**Algorithm 2** Data Generation for Topological Jigsaw Reconstruction

---

**Require:** Original Image $I \in \mathbb{R}^{H \times W}$
**Require: Hyperparameters:**
 1:    Grid Dimension $G \leftarrow 2$ {Partitions image into $2 \times 2$ quadrants}
 2:    Patch Height $H_{\text{p}} \leftarrow H//G$
 3:    Patch Width $W_{\text{p}} \leftarrow W//G$
 4:    Canonical Indices $\mathcal{K} \leftarrow \{0, 1, 2, 3\}$ {Reading order: TL, TR, BL, BR}
**Ensure:** Shuffled Image $I_{\text{shuffled}}$, Target Sequence $Y^*$
 5: // *Step 1: Extract Canonical Patches*
 6: Define grid coordinates $C \leftarrow \{(0, 0), (0, W_{\text{p}}), (H_{\text{p}}, 0), (H_{\text{p}}, W_{\text{p}})\}$
 7: Initialize patch library $\mathcal{P}_{\text{lib}} \leftarrow [\,]$
 8: **for** $k \in \mathcal{K}$ **do**
 9:    $(y, x) \leftarrow C[k]$
10:    Extract $P \leftarrow I[y : y + H_{\text{p}}, \ x : x + W_{\text{p}}]$
11:    Append $P$ to $\mathcal{P}_{\text{lib}}$
12: **end for**
13: // *Step 2: Generate Permutation*
14: Generate random permutation $\sigma$ of $\mathcal{K}$
15: Initialize canvas $I_{\text{shuffled}}$ of size $H \times W$
16: // *Step 3: Reconstruct with Shuffled Order*
17: **for** $i \in \mathcal{K}$ **do**
18:    $idx_{\text{source}} \leftarrow \sigma[i]$ {Select which original patch goes to position $i$}
19:    $(y_{\text{target}}, x_{\text{target}}) \leftarrow C[i]$ {Get coordinates for position $i$}
20:    Place $\mathcal{P}_{\text{lib}}[idx_{\text{source}}]$ into $I_{\text{shuffled}}$ at $(y_{\text{target}}, x_{\text{target}})$
21: **end for**
22: // *Target is the permutation sequence defining the layout*
23: $Y^* \leftarrow \sigma$
24: **return** $I_{\text{shuffled}}, Y^*$

---

### A.2.3. ANOMALY CONSISTENCY DETECTION

This task trains the model to detect subtle anomalies and structural errors. We generate these examples using a "cut-paste" method: a section of the anatomy is replaced by a similar-looking, but incorrect, patch from a reference image ($I_{\text{ref}}$). To prevent the model from cheating by detecting sharp edges, we use Gaussian noise to blend the boundaries where the images meet.

**Reference Image Selection ($I_{\text{ref}}$).** To ensure the anomaly is non-trivial, we select $I_{\text{ref}}$ based on modality-specific hardness:

- **Volumetric (CT/MRI):** $I_{\text{ref}}$ is selected from a nearby slice ($z \pm 5$). This ensures the overall organ shape remains consistent, while introducing slight natural variations.

- **Planar (X-ray):** $I_{\text{ref}}$ is the top-1 retrieval from the dataset via BiomedCLIP embedding similarity, ensuring semantic density consistency.

**Generation Protocol.** We partition the image into a $4 \times 4$ grid. The anomaly is injected into the central region defined by $\mathcal{C}_{\text{center}}$. Crucially, to mitigate the "sharp edge" artifact common in cut-paste augmentations, we add Gaussian noise to the boundary pixels of the pasted region. The detailed procedure and hyperparameters are formalized in Algorithm 3.

---

**Algorithm 3** Data Generation for Anomaly Consistency Detection

---

**Require:** Target Image $I \in \mathbb{R}^{H \times W}$, Modality $M$, Database $\mathcal{D}$
**Require: Hyperparameters:**
  1:     Grid Dimension $G \leftarrow 4$
  2:     Patch Size $S \leftarrow (H/G, W/G)$
  3:     Center Indices $\mathcal{C}_{\text{center}} \leftarrow \{5, 6, 9, 10\}$ {Central $2 \times 2$ block in flattened index}
  4:     Noise Level $\sigma_{\text{noise}} \leftarrow 0.05$ {Std dev for boundary blending}
  5:     Boundary Width $\delta \leftarrow 2$ {Pixel width for noise injection}
**Ensure:** Anomalous Image $I_{\text{anom}}$, Anomaly Index $k^*$
  6:  *// Step 1: Select Hard Negative Reference*
  7:  **if** $M \in \{\text{CT}, \text{MRI}\}$ **then**
  8:     $I_{\text{ref}} \leftarrow \text{GetSlice}(I.volume, I.z \pm 1)$
  9:  **else**
10:     $v_{\text{I}} \leftarrow \text{BioMedCLIP}(I)$
11:     $I_{\text{ref}} \leftarrow \text{argmax}_{\text{J} \in \mathcal{D}, \text{J} \neq \text{I}}(\text{CosSim}(v_{\text{I}}, v_{\text{J}}))$
12:  **end if**
13:  *// Step 2: Inject Anomaly with Edge Blending*
14:  Partition $I_{\text{ref}}$ into grid $\mathcal{P}_{\text{ref}}$
15:  Sample target index $k^* \sim \text{Uniform}(\mathcal{C}_{\text{center}})$
16:  Extract foreign patch $P_{\text{foreign}} \leftarrow \mathcal{P}_{\text{ref}}[k^*]$
17:  $I_{\text{anom}} \leftarrow I$
18:  Paste $P_{\text{foreign}}$ into $I_{\text{anom}}$ at position $k^*$
19:  *// Step 3: Apply Boundary Noise*
20:  Get boundary region $\Omega$ of width $\delta$ around position $k^*$
21:  Generate noise $\epsilon \sim \mathcal{N}(0, \sigma_{\text{noise}})$
22:  $I_{\text{anom}}[\Omega] \leftarrow I_{\text{anom}}[\Omega] + \epsilon$
23:  **return** $I_{\text{anom}}, k^*$

---

### A.3. Unified VQA Instruction Formatting

To facilitate end-to-end training using a unified objective, we standardize all geometric proxy tasks into a consistent open-set VQA format. Instead of using task-specific heads, we formulate these tasks as natural language conversations.

As illustrated in Figure 9, each training instance is composed of three standardized components:

1. **Visual Input:** We use special tokens (e.g., `<image>`) to represent the medical scans. Note that for the *Hierarchical Scale Localization* task (Figure 9a), the input specifically supports multi-image sequences (Global View + Local Crops).

2. **User Prompt:** A structured instruction that clearly defines the geometric objective and constrains the output format.

3. **Target Response:** To support diverse inference strategies, we define two distinct output states:

   - **Direct Mode:** The model directly outputs the concise final answer (e.g., sequence indices or grid coordinates), focusing on strict format.
   - **Reasoning Mode:** The model first generates a CoT reasoning path enclosed in `<think>...</think>` tags to articulate geometric constraints before deriving the final answer enclosed in `<answer>...</answer>` tags.

Visual examples of the direct mode and the reasoning mode are provided in Figure 9 and Figure 10.

## Task A: Hierarchical Scale Localization

<image><image><image><image>The first image provided is the original CT scan. The following three images are patches cropped from it at different scales and resized to the original dimensions.\n\nFor each cropped patch, classify its scale relative to the original image:\n- Assign 'Level 1' if the patch covers approximately 20% of the original area (half dimension).\n- Assign 'Level 2' if the patch covers approximately 6.25% of the original area (quarter dimension).\n\nPlease output the label for each patch in order, separated by commas. For example: \"Level 1, Level 2, Level 1\".

Level 1, Level 2, Level 2

## Task B: Topological Jigsaw Reconstruction

<image>This is a MR image that has been divided into a 2x2 grid and shuffled. Please identify the index position (1-4) of each patch in the shuffled image according to its original position, in left-to-right, top-to-bottom order. For example, \"1 3 2 4\".

1 3 4 2

## Task C: Anomaly Consistency Detection

<image>This is a X-ray image that has been divided into a 4x4 grid. 1 patch in this image has been replaced with a patch from another image. Please identify which patch appears anomalous and output its position number (numbered 1-16 from left to right, top to bottom). For example, \"8\".

7

*Figure 9.* Unified VQA Instruction Examples of Direct Mode.

### A.4. Med-Scout-Bench Evaluation Pipeline

To ensure consistent scoring across models with different styles, we use a standardized evaluation process as shown in Figure 11. Since MLLMs often provide detailed reasoning that makes simple text matching difficult, we use DeepSeek-V3.2 (DeepSeek-AI et al., 2025) to extract the conclusion from the raw output. These extracted answers are then automatically compared against the correct labels using a strict scoring script.

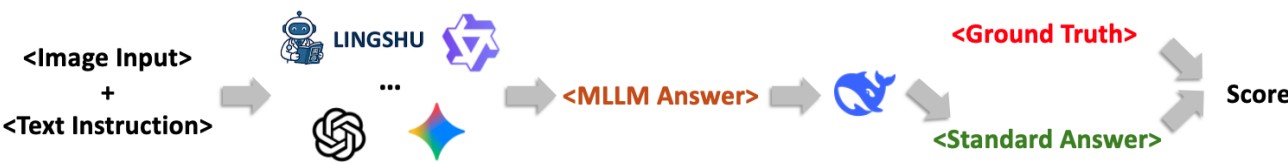

*Figure 11.* Evaluation pipeline on Med-Scout-Bench.

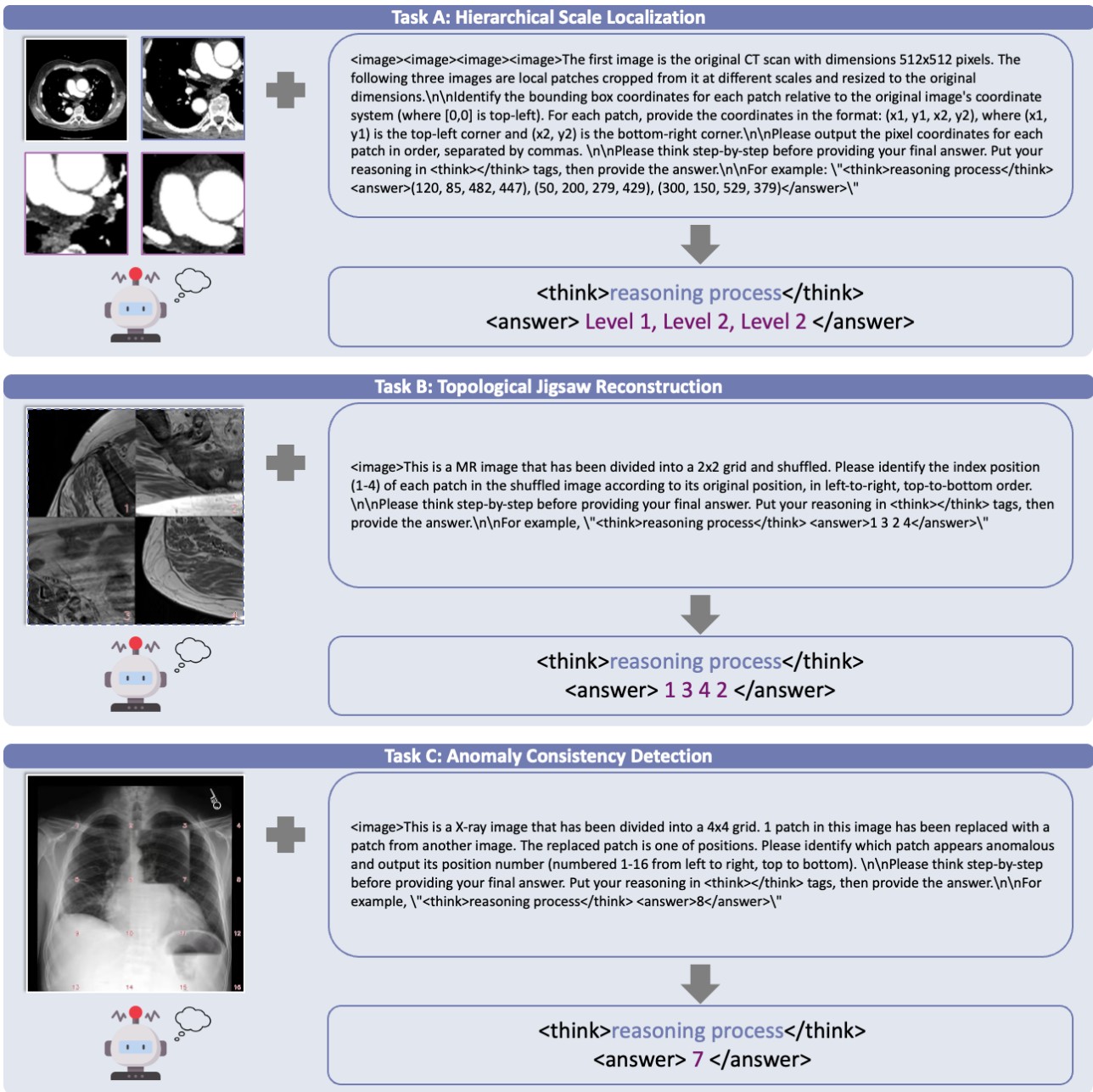

*Figure 10.* Unified VQA Instruction Examples of Reasoning Mode.

## B. Training Implementation Details

### B.1. Hyperparameters

We provide a comprehensive list of hyperparameters used during the Med-Scout RL post-training phase in Table 5. The training configuration is standardized across all backbone models to ensure fair comparison.

### B.2. Reward Curves

To verify the stability and convergence of our geometry-aware post-training, we visualize the reward trajectories during the GRPO training phase. Figure 12 presents the comprehensive learning curves for both the Direct Mode and the Reasoning

*Table 5.* Comprehensive list of hyperparameters for Med-Scout training. The configuration covers optimization, GRPO strategy, reward engineering, and system settings.

| HYPERPARAMETER | VALUE | DESCRIPTION |
|---|---|---|
| *Optimization Configuration* | | |
| OPTIMIZER | ADAMW | $\beta_1 = 0.9, \beta_2 = 0.95$ |
| PEAK LEARNING RATE | $1 \times 10^{-6}$ | LOWER THAN SFT TO PREVENT COLLAPSE |
| LR SCHEDULER | COSINE DECAY | MINIMUM LR SET TO $1 \times 10^{-7}$ |
| WARM-UP RATIO | 0.01 | LINEAR WARM-UP STRATEGY |
| WEIGHT DECAY | 0.1 | STANDARD REGULARIZATION |
| TRAINING STEPS | 7200 | TOTAL OPTIMIZATION UPDATES |
| *GRPO Strategy* | | |
| GLOBAL BATCH SIZE | 192 | 192 FOR ALL MODELS |
| GROUP SIZE ($G$) | 8 | NUMBER OF OUTPUTS SAMPLED PER PROMPT |
| KL COEFFICIENT ($\beta$) | 0.04 | PENALTY WEIGHT FOR POLICY DRIFT |
| CLIP RATIO ($\epsilon$) | 0.2 | STANDARD PPO CLIPPING RANGE |
| *Reward Engineering* | | |
| TOTAL MAX REWARD | 2.0 | SUM OF ACCURACY, FORMAT, AND REASONING REWARDS |
| ACCURACY CAP ($\mathcal{R}_{ACC}$) | 1.0 | TASK-SPECIFIC GEOMETRIC PRECISION |
| FORMAT CAP ($\mathcal{R}_{FMT}$) | 0.5 | SYNTAX COMPLIANCE REWARD |
| REASONING CAP ($\mathcal{R}_{REASON}$) | 0.5 | ACTIVE ONLY IN REASONING MODE (COT) |
| ANOMALY TEMP ($\tau$) | 0.1 | TEMPERATURE FOR DISTANCE-BASED REWARD |
| *System & Generation* | | |
| MAX NEW TOKENS | 1024 | BUFFER FOR COT REASONING TRACES |
| PRECISION | BF16 | MIXED PRECISION TRAINING |
| HARDWARE | 6× RTX PRO 6000 | NVIDIA RTX PRO 6000 GPUs |

Mode across the four backbone models.

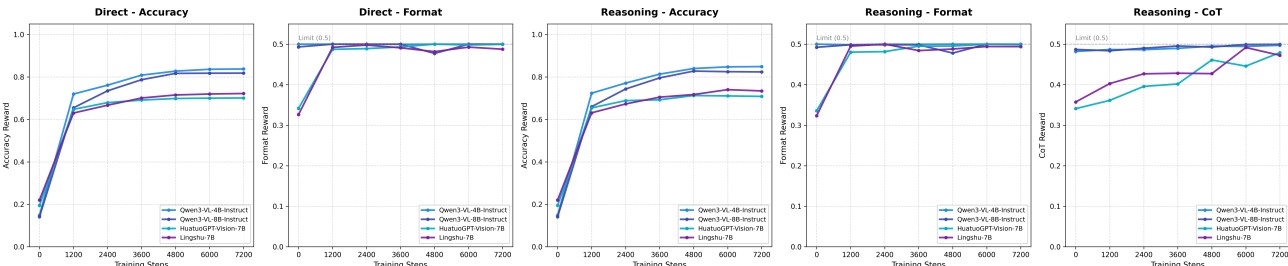

*Figure 12.* Training Reward Dynamics. The plots illustrate the optimization trajectories for Direct Mode (left two panels) and Reasoning Mode (right three panels). We report the Dense Geometric Rewards ($\mathcal{R}_{acc}$), Format Rewards ($\mathcal{R}_{fmt}$), and Reasoning Structure Rewards ($\mathcal{R}_{reason}$).

### B.2.1. DIRECT MODE DYNAMICS

The first two panels of Figure 12 illustrate the training dynamics under the Direct Mode setting.

- **Dense Geometric Reward** ($\mathcal{R}_{\mathbf{acc}}$): The *Direct - Accuracy* subplot depicts the steady improvement in geometric perception. We observe that general-domain models (Qwen3-VL series) achieve slightly higher accuracy ($\sim 0.84$) compared to medical specialists ($\sim 0.72$). This suggests that strong foundational vision-language capabilities are beneficial for solving complex spatial reasoning tasks.

- **Format Reward** ($\mathcal{R}_{\mathbf{fmt}}$): As shown in the *Direct - Format* subplot, all models rapidly master the requisite output format constraints (e.g., coordinate normalization, comma-separated lists). The format reward converges to the maximum value of 0.5 within the first 1,200 steps, indicating that the syntax of the geometric proxy tasks is easily learnable.

### B.2.2. REASONING MODE DYNAMICS

In the Reasoning Mode, models are required to generate a CoT trace before the final answer. The last three panels of Figure 12 display the corresponding reward curves.

- **Dense Geometric Reward ($\mathcal{R}_{\mathbf{acc}}$):** Compared to Direct Mode, the Reasoning Mode typically exhibits a slightly slower convergence rate initially due to the increased generation length and complexity. However, the final convergence values remain consistent with the Direct Mode, validating the robustness of the alignment strategy.

- **Format Reward ($\mathcal{R}_{\mathbf{fmt}}$):** Similar to the Direct Mode, the format compliance in Reasoning Mode exhibits extremely rapid convergence. As shown in the *Reasoning - Format* subplot, all models quickly saturate at the maximum reward of 0.5, confirming that the increased sequence length from the reasoning process does not degrade the model's ability to adhere to syntactic instructions.

- **Reasoning Structure Reward ($\mathcal{R}_{\mathbf{reason}}$):** As illustrated in the *Reasoning - CoT* subplot, the models effectively adapt to the `<think>...</think>` and `<answer>...</answer>` structure. Notably, while the general-domain Qwen3 models maintain high structural adherence from the start, the medical specialists (HuatuoGPT, Lingshu) show a distinct learning curve, requiring approximately 3,600 steps to converge to the maximum structural reward.

## C. Extensive Experimental Results

### C.1. Detailed Evaluation on Med-Scout-Bench

Due to space constraints in the main text, we focused on the aggregated geometric alignment scores. In this section, we provide a comprehensive performance breakdown for each backbone model across the three specific proxy tasks: Hierarchical Scale Localization ($\mathcal{T}_{\text{scale}}$), Topological Jigsaw Reconstruction ($\mathcal{T}_{\text{topo}}$), and Anomaly Consistency Detection ($\mathcal{T}_{\text{anom}}$).

The detailed numerical results are presented in Table 6. Beyond the performance gains, three notable patterns emerge from our findings:

- **General-Domain Models Learn Faster and Better:** General-purpose models (e.g., Qwen3-VL series) consistently outperform medical specialist models (e.g., Lingshu, HuatuoGPT) in learning geometric concepts. As shown in the reward curves and final scores, general models converge more rapidly and achieve significantly higher final accuracy. This suggests that a strong fundamental vision-language capability is more critical for grasping abstract spatial logic than domain-specific medical training.

- **Smaller Models Can Be More Efficient (4B vs. 8B):** Contrary to the scaling law expectation that "larger is better," we observe that the Qwen3-VL-4B model learns faster and achieves higher geometric accuracy (Avg. 84.4) compared to its larger 8B counterpart (Avg. 83.6). This indicates that for targeted geometric alignment, parameter efficiency and training dynamics may matter more than sheer model size.

- **Med-Scout Bridges the Gap with Proprietary Models:** In their base states, proprietary models like GPT-5 and Gemini-3-Flash significantly outperform open-source models (scoring 60% vs. 30%), likely due to their superior pre-training. However, after Med-Scout post-training, open-source models not only close this gap but easily surpass these closed-source leaders, with scores jumping to the 70%-90% range. This demonstrates that specific geometric supervision is highly effective at unlocking capabilities that even the strongest proprietary models lack.

### C.2. Evaluation on Report Generation Tasks

The comprehensive results on the MIMIC-CXR and IU-Xray benchmarks are listed in Table 3. Beyond improving general-purpose models, Med-Scout proves effective at elevating already powerful domain specialists. Lingshu-7B, previously established as a SOTA medical MLLM, breaks through its performance ceiling after our alignment. It achieves a new SOTA CIDEr score on MIMIC-CXR, significantly outperforming proprietary commercial models. This indicates that our method provides a complementary geometric capability that is missing even in extensively trained medical models.

*Table 6.* Performance evaluation on the Med-Scout-Bench. The table reports scores on three distinct subtasks and their average. All scores are scaled by a factor of 100 for better readability.

| MODEL | MED-SCOUT-BENCH | | | |
|---|---|---|---|---|
| | TASK A | TASK B | TASK C | AVG. |
| *Proprietary Models* | | | | |
| GPT-5 | 64.3 | 56.1 | 74.6 | 63.6 |
| GEMINI-3-FLASH | 67.0 | 58.8 | 76.7 | 66.1 |
| *General-purpose MLLMs* | | | | |
| INTERNVL3-8B | 37.7 | 34.3 | 27.3 | 32.5 |
| QWEN2.5-VL-3B-INSTRUCT | 46.9 | 34.6 | 9.2 | 28.2 |
| QWEN2.5-VL-7B-INSTRUCT | 31.4 | 28.1 | 27.8 | 28.5 |
| QWEN2.5-VL-32B-INSTRUCT | 28.9 | 37.3 | 19.6 | 30.0 |
| QWEN2.5-VL-72B-INSTRUCT | 32.7 | 29.9 | 30.4 | 30.5 |
| QWEN3-VL-4B-INSTRUCT | 59.6 | 36.6 | 31.3 | 38.7 |
| + MED-SCOUT | 94.4↑34.8 | 77.5↑40.9 | 89.8↑58.5 | 84.4↑45.7 |
| QWEN3-VL-8B-INSTRUCT | 41.0 | 34.7 | 46.4 | 39.7 |
| + MED-SCOUT | 86.7↑45.7 | 78.1↑43.4 | 90.2↑43.8 | 83.6↑43.9 |
| QWEN3-VL-32B-INSTRUCT | 43.7 | 36.2 | 48.6 | 41.6 |
| *Medical MLLMs* | | | | |
| LLAVA-MED-7B | 21.9 | 17.4 | 8.6 | 15.2 |
| MEDGEMMA-4B-IT | 46.4 | 33.5 | 29.0 | 34.1 |
| MEDGEMMA-27B-IT | 47.3 | 30.2 | 33.4 | 34.1 |
| HUATUOGPT-VISION-7B | 60.7 | 33.7 | 28.0 | 36.3 |
| + MED-SCOUT | 76.6↑15.9 | 79.3↑45.6 | 64.0↑36.0 | 73.8↑37.5 |
| HUATUOGPT-VISION-34B | 62.9 | 33.9 | 30.8 | 37.7 |
| LINGSHU-7B | 60.6 | 29.1 | 21.9 | 31.9 |
| + MED-SCOUT | 78.9↑18.3 | 77.5↑48.4 | 60.0↑38.1 | 71.9↑40.0 |
| LINGSHU-32B | 63.6 | 31.8 | 20.4 | 33.3 |

## C.3. Task Difficulty Analysis

To investigate whether the complexity of the geometric proxy tasks contributes to the final post-training performance, we designed a controlled ablation study with three difficulty levels using Direct Mode. We constructed "Easy" and "Medium" variants of the training dataset by adjusting the complexity of the proxy tasks:

- **Easy Variant:**
    - *Scale:* Single crop input ($N = 1$), removing the need for multi-scale comparative reasoning.
    - *Topology:* Simple $1 \times 2$ grid shuffling, requiring only binary relative positioning.
    - *Anomaly:* Coarse $2 \times 2$ grid (4 patches), making the foreign patch visually prominent.

- **Medium Variant:**
    - *Scale:* Two crop inputs ($N = 2$), introducing limited multi-view context.
    - *Topology:* $1 \times 4$ linear strip shuffling, increasing sequence length but lacking vertical spatial logic.
    - *Anomaly:* Intermediate $4 \times 2$ grid (8 patches), requiring moderate attention granularity.

- **Hard Variant (Med-Scout Standard):**
    - *Scale:* Three hierarchical crops ($N = 3$), forcing robust global-local mapping.
    - *Topology:* $2 \times 2$ grid shuffling, necessitating 2D spatial reasoning (both horizontal and vertical).
    - *Anomaly:* Fine-grained $4 \times 4$ grid (16 patches) with hard-negative mining, requiring pixel-level scrutiny.

**Results.** As shown in Table 7, while the difficulty level increases from "Easy" variants to the "Hard" (standard Med-Scout) setting, the model consistently achieves superior accuracy across external benchmarks. The "Hard" configuration, which enforces rigorous constraints, proved essential for achieving the best results. This confirms that high-complexity geometric objectives are necessary to prevent models from relying on superficial pattern matching, instead compelling them to master deep, pixel-level visual reasoning.

*Table 7.* Impact of Training Task Difficulty. We evaluate the Qwen3-VL-8B-Instruct model trained with datasets of varying geometric complexity (Easy, Medium, Hard) across radiological and general medical VQA benchmarks. Higher difficulty in proxy tasks consistently leads to superior generalization performance. The best results are highlighted.

| TRAINING DIFFICULTY | RADIOLOGICAL VQA | | | GENERALIZATION | | |
|---|---|---|---|---|---|---|
| | RAD-VQA | VQA-RAD | SLAKE | PMC-VQA | OMNIMEDVQA | MEDXPERTQA |
| *Baseline* | | | | | | |
| QWEN3-VL-8B-INSTRUCT | 41.6 | 63.2 | 69.6 | 43.9 | 42.9 | 30.4 |
| *Med-Scout Variants* | | | | | | |
| EASY DIFFICULTY | 43.9↑2.3 | 65.1↑1.9 | 71.4↑1.8 | 44.2↑0.3 | 45.1↑2.2 | 30.5↑0.1 |
| MEDIUM DIFFICULTY | 44.8↑3.2 | 64.7↑1.5 | 71.8↑2.2 | 45.0↑1.1 | 45.7↑2.8 | 30.5↑0.1 |
| **HARD DIFFICULTY (OURS)** | **45.3**↑3.7 | **65.8**↑2.6 | **72.0**↑2.4 | **45.5**↑1.6 | **46.0**↑3.1 | **30.8**↑0.4 |
| Δ | +0.5 | +0.7 | +0.2 | +0.5 | +0.3 | +0.3 |

*Table 8.* Ablation Study of Proxy Task. We use Qwen3-VL-8B-Instruct as the backbone model to evaluate how different geometric tasks contribute to generalization performance across radiological and broad medical domains.

| TRAINING CONFIGURATION | RADIOLOGICAL VQA | | | GENERALIZATION | | |
|---|---|---|---|---|---|---|
| | RAD-VQA | VQA-RAD | SLAKE | PMC-VQA | OMNIMEDVQA | MEDXPERTQA |
| *Baseline* | | | | | | |
| QWEN3-VL-8B-INSTRUCT | 41.6 | 63.2 | 69.6 | 43.9 | 42.9 | 30.4 |
| *Single Task Specialists* | | | | | | |
| + SCALE ONLY | 42.9↑1.3 | 64.1↑0.9 | 70.3↑0.7 | 44.1↑0.2 | 43.9↑1.0 | 30.5↑0.1 |
| + TOPOLOGY ONLY | 44.1↑2.5 | 64.9↑1.7 | 70.6↑1.0 | 44.8↑0.9 | 44.5↑1.6 | 30.7↑0.3 |
| + ANOMALY ONLY | 44.6↑3.0 | 64.9↑1.7 | 71.0↑1.4 | 44.4↑0.5 | 44.7↑1.8 | 30.6↑0.2 |
| *Leave-One-Out* | | | | | | |
| + MED-SCOUT (W/O SCALE) | 44.9↑3.3 | 65.1↑1.9 | 71.4↑1.8 | 44.8↑0.9 | 45.3↑2.4 | 30.6↑0.2 |
| + MED-SCOUT (W/O TOPOLOGY) | 44.7↑3.1 | 65.0↑1.8 | 71.1↑1.5 | 45.0↑1.1 | 45.5↑2.6 | 30.5↑0.1 |
| + MED-SCOUT (W/O ANOMALY) | 44.8↑3.2 | 64.8↑1.6 | 70.8↑1.2 | 45.2↑1.3 | 45.4↑2.5 | 30.7↑0.3 |
| **+ MED-SCOUT (FULL)** | **45.3**↑3.7 | **65.8**↑2.6 | **72.0**↑2.4 | **45.5**↑1.6 | **46.0**↑3.1 | **30.8**↑0.4 |
| Δ | +0.4 | +0.7 | +0.6 | +0.3 | +0.5 | +0.1 |

## C.4. Impact of Proxy Task Types

To investigate the distinct contribution of each geometric proxy task to the overall performance, we conducted a comprehensive ablation study. Using Qwen3-VL-8B-Instruct as the backbone, we compared the baseline performance against variants trained with *Single Task Specialists* (using only one task type), *Leave-One-Out* Configurations (removing exactly one task type), and the full Med-Scout framework across six benchmarks.

The results are reported in Table 8. We observe that:

• **Every Proxy Task is Indispensable.** Comparing the full Med-Scout framework against the "Leave-One-Out" configurations reveals that removing any single geometric task leads to a consistent performance degradation across all benchmarks. This confirms that these three proxy tasks collectively establishing a holistic geometric perception that is superior to the sum of its parts.

• **Topology and Anomaly Tasks Drive Generalization.** The latter two tasks Topological Jigsaw Reconstruction ($\mathcal{T}_{\text{topo}}$) and Anomaly Consistency Detection ($\mathcal{T}_{\text{anom}}$) demonstrate a more critical impact on model performance and generalization. This suggests that the high-level logical deduction required for topology and the fine-grained scrutiny needed for anomaly detection are fundamental capabilities that generalize effectively to diverse medical imaging modalities.

## C.5. Impact of Explicitly Rewarding Logical Validity in CoT

To investigate if explicitly rewarding the logical validity of the CoT reasoning process could further enhance the model's geometric perception capabilities, we conducted an ablation study comparing three distinct post-training configurations

across multiple benchmarks:

- **(M)**: Med-Scout Direct Mode.

- **(MR)**: Med-Scout Reasoning Mode (enforcing the `<think>...</think>` format).

- **(MRL)**: Med-Scout Reasoning Mode with an additional explicit reward tailored to evaluate the logical validity of the reasoning steps.

As shown in Table 9, incorporating an explicit logic reward fails to yield overall performance growth and occasionally underperforms compared to the direct and standard reasoning modes. This underperformance suggests that enforcing explicit textual reasoning offers very limited benefits for strong visual tasks that fundamentally rely on rigorous, low-level geometric perception rather than semantic logic.

*Table 9.* Performance comparison of explicitly rewarding the logical validity of the CoT process. (M) denotes Direct Mode, (MR) denotes Reasoning Mode with structural reward, and (MRL) denotes Reasoning Mode with an additional explicit logical validity reward. All scores are scaled by a factor of 100.

| MODEL | RAD-VQA | VQA-RAD | SLAKE | PMC-VQA | OMNIMEDVQA | MEDXPERTQA |
|---|---|---|---|---|---|---|
| **QWEN3-VL-8B-INSTRUCT** | 41.6 | 63.2 | 69.6 | 43.9 | 42.9 | 30.4 |
| (M) | 45.3↑3.7 | 65.8↑2.6 | 72.0↑2.4 | 45.5↑1.6 | 46.0↑3.1 | 30.8↑0.4 |
| (MR) | 45.9↑4.3 | 66.4↑3.2 | 73.3↑3.7 | 44.4↑0.5 | 46.9↑4.0 | 30.7↑0.3 |
| (MRL) | 45.2↑3.6 | 66.1↑2.9 | 72.7↑3.1 | 45.0↑1.1 | 46.6↑3.7 | 30.8↑0.4 |
| **HUATUOGPT-VISION-7B** | 48.8 | 67.0 | 67.8 | 53.0 | 75.0 | 22.4 |
| (M) | 52.1↑3.3 | 70.1↑3.1 | 71.0↑3.2 | 55.9↑2.9 | 75.4↑0.4 | 22.7↑0.3 |
| (MR) | 53.0↑4.2 | 70.3↑3.3 | 69.8↑2.0 | 56.8↑3.8 | 76.2↑1.2 | 22.7↑0.3 |
| (MRL) | 53.4↑4.6 | 70.1↑3.1 | 71.0↑3.2 | 57.3↑4.3 | 75.8↑0.8 | 22.4↑0.0 |
| **LINGSHU-7B** | 61.2 | 68.9 | 82.8 | 56.3 | 81.4 | 27.4 |
| (M) | 64.0↑2.8 | 71.0↑2.1 | 83.0↑0.2 | 57.4↑1.1 | 81.9↑0.5 | 28.0↑0.6 |
| (MR) | 63.8↑2.6 | 70.8↑1.9 | 83.0↑0.2 | 57.6↑1.3 | 81.6↑0.2 | 28.5↑1.1 |
| (MRL) | 64.0↑2.8 | 70.3↑1.4 | 83.6↑0.8 | 57.8↑1.5 | 81.9↑0.5 | 28.7↑1.3 |

## C.6. Reward Mechanism Comparison

A critical component of the Med-Scout framework is the DGR mechanism, designed to overcome the sparsity of binary feedback in complex reasoning tasks. To quantify its impact, we compared our approach against a standard sparse reward baseline.

- **Sparse Reward Setting:** The model receives a reward of $\mathcal{R} = 1$ only if the generated answer perfectly matches the ground truth (e.g., exact index sequence or coordinates within a strict threshold); otherwise, $\mathcal{R} = 0$.

- **Dense Reward Setting (Ours):** As detailed in Section 4.2, we utilize continuous metrics including IoU for bounding boxes, Euclidean distance decay for anomaly detection, and element-wise alignment for topological sequences.

Table 10 presents the comparison using the Qwen3-VL-4B-Instruct and Qwen3-VL-8B-Instruct backbones. The results demonstrate that dense geometric reward provides a distinct optimization advantage over binary feedback. While the sparse reward setting already yields notable improvements over the baseline, the DGR mechanism consistently outperforms the sparse variant across all six benchmarks.

This confirms that the granular feedback provided by DGR is crucial for efficient RL post-training. Specifically, DGR awards partial credit for outputs such as "near-miss" scale predictions or approximate anomaly locations. In contrast, rigid pass/fail signals often fail to provide useful gradients for partially correct reasoning. Our DGR effectively guides the model to progressively refine its geometric understanding, which leads to superior generalization across both radiological and broad medical domains.

## C.7. SFT vs. RL

We observe a distinct contrast between internal alignment scores and external generalization capabilities:

*Table 10.* Impact of Reward Mechanism. Performance comparison between standard sparse reward and our dense geometric reward on six benchmarks. The dense mechanism provides granular feedback, leading to significantly better generalization.

| MODEL & REWARD STRATEGY | RADIOLOGICAL VQA | | | GENERALIZATION | | |
|---|---|---|---|---|---|---|
| | RAD-VQA | VQA-RAD | SLAKE | PMC-VQA | OMNIMEDVQA | MEDXPERTQA |
| **QWEN3-VL-4B-INSTRUCT** | | | | | | |
| BASELINE | 41.5 | 59.9 | 73.4 | 42.8 | 45.5 | 27.0 |
| W/ SPARSE REWARD | 45.1↑3.6 | 62.0↑2.1 | 75.3↑1.9 | 44.7↑1.9 | 48.6↑3.1 | 27.3↑0.3 |
| **W/ DGR (OURS)** | **45.7**↑4.2 | **62.9**↑3.0 | **75.6**↑2.2 | **45.1**↑2.3 | **48.8**↑3.3 | **27.7**↑0.7 |
| Δ (DGR VS SPARSE) | +0.6 | +0.9 | +0.3 | +0.4 | +0.2 | +0.4 |
| **QWEN3-VL-8B-INSTRUCT** | | | | | | |
| BASELINE | 41.6 | 63.2 | 69.6 | 43.9 | 42.9 | 30.4 |
| W/ SPARSE REWARD | 44.7↑3.1 | 65.2↑2.0 | 71.3↑1.7 | 45.0↑1.1 | 45.8↑2.9 | 30.8↑0.4 |
| **W/ DGR (OURS)** | **45.3**↑3.7 | **65.8**↑2.6 | **72.0**↑2.4 | **45.5**↑1.6 | **46.0**↑3.1 | **30.8**↑0.4 |
| Δ (DGR VS SPARSE) | +0.6 | +0.6 | +0.7 | +0.5 | +0.2 | +0.0 |

*Table 11.* Comparison of SFT vs. RL on Med-Scout-Bench.

| MODEL / METHOD | MED-SCOUT-BENCH | | | |
|---|---|---|---|---|
| | TASK A | TASK B | TASK C | AVG. |
| *Qwen3-VL-4B-Instruct* | | | | |
| BASELINE | 59.6 | 36.6 | 31.3 | 38.7 |
| + MED-SCOUT (SFT) | 89.9↑30.3 | **78.1**↑41.5 | 84.6↑53.3 | 82.2↑43.5 |
| **+ MED-SCOUT (RL)** | **94.4**↑34.8 | 77.5↑40.9 | **89.8**↑58.5 | **84.4**↑45.7 |
| Δ | +4.5 | -0.6 | +5.2 | +2.2 |
| *Qwen3-VL-8B-Instruct* | | | | |
| BASELINE | 41.0 | 34.7 | 46.4 | 39.7 |
| + MED-SCOUT (SFT) | 85.1↑44.1 | 76.9↑42.2 | 88.7↑42.3 | 82.2↑42.5 |
| **+ MED-SCOUT (RL)** | **86.7**↑45.7 | **78.1**↑43.4 | **90.2**↑43.8 | **83.6**↑43.9 |
| Δ | +1.6 | +1.2 | +1.5 | +1.4 |
| *HuatuoGPT-Vision-7B* | | | | |
| BASELINE | 60.7 | 33.7 | 28.0 | 36.3 |
| + MED-SCOUT (SFT) | **79.2**↑18.5 | 79.3↑45.6 | **65.1**↑37.1 | **74.6**↑38.3 |
| **+ MED-SCOUT (RL)** | 76.6↑15.9 | 79.3↑45.6 | 64.0↑36.0 | 73.8↑37.5 |
| Δ | -2.6 | +0.0 | -1.1 | -0.8 |
| *Lingshu-7B* | | | | |
| BASELINE | 60.6 | 29.1 | 21.9 | 31.9 |
| + MED-SCOUT (SFT) | 72.8↑12.2 | **78.5**↑49.4 | 58.0↑36.1 | 70.7↑38.8 |
| **+ MED-SCOUT (RL)** | **78.9**↑18.3 | 77.5↑48.4 | **60.0**↑38.1 | **71.9**↑40.0 |
| Δ | +6.1 | -1.0 | +2.0 | +1.2 |

- **SFT Achieves Strong Performance on Internal Validation.** On the internal Med-Scout-Bench (Table 11), SFT demonstrates remarkable efficacy. It achieves performance levels comparable to or even surpassing the RL-tuned models. For example, HuatuoGPT-Vision-7B achieves an average score of 74.6% with SFT, compared to 73.8% with RL. This indicates that models can easily master the output syntax and specific data patterns of the proxy tasks through imitation.

- **RL Enables True Generalization.** However, the apparent competence of SFT collapses on external benchmarks (Table 12). SFT variants exhibit negligible or even negative performance shifts. For instance, Qwen3-VL-8B-Instruct drops by 0.7% on Rad-VQA and 0.3% on MedXpertQA. This reveals that SFT merely overfits to the proxy task patterns without internalizing the underlying geometric reasoning. In contrast, RL achieves consistent gains across all external benchmarks. This confirms that exploration-driven optimization is essential for cultivating a generalized geometric perception that transfers beyond the training data.

*Table 12.* Comparison of SFT vs. RL on six external benchmarks.

| Model / Method | Radiological VQA | | | Generalization | | |
| --- | --- | --- | --- | --- | --- | --- |
| | Rad-VQA | VQA-RAD | SLAKE | PMC-VQA | OmniMedVQA | MedXpertQA |
| *Qwen3-VL-4B-Instruct* | | | | | | |
| Baseline | 41.5 | 59.9 | 73.4 | 42.8 | 45.5 | 27.0 |
| + Med-Scout (SFT) | 41.6↑0.1 | 59.3↓0.6 | 73.3↓0.1 | 42.8↑0.0 | 45.7↑0.2 | 26.8↓0.2 |
| **+ Med-Scout (RL)** | **45.7**↑4.2 | **62.9**↑3.0 | **75.6**↑2.2 | **45.1**↑2.3 | **48.8**↑3.3 | **27.7**↑0.7 |
| Δ | +4.1 | +3.6 | +2.3 | +2.3 | +3.1 | +0.9 |
| *Qwen3-VL-8B-Instruct* | | | | | | |
| Baseline | 41.6 | 63.2 | 69.6 | 43.9 | 42.9 | 30.4 |
| + Med-Scout (SFT) | 40.9↓0.7 | 63.1↓0.1 | 69.8↑0.2 | 44.5↑0.6 | 42.7↓0.2 | 30.1↓0.3 |
| **+ Med-Scout (RL)** | **45.3**↑3.7 | **65.8**↑2.6 | **72.0**↑2.4 | **45.5**↑1.6 | **46.0**↑3.1 | **30.8**↑0.4 |
| Δ | +4.4 | +2.7 | +2.2 | +1.0 | +3.3 | +0.7 |
| *HuatuoGPT-Vision-7B* | | | | | | |
| Baseline | 48.8 | 67.0 | 67.8 | 53.0 | 75.0 | 22.4 |
| + Med-Scout (SFT) | 48.5↓0.3 | 67.1↑0.1 | 68.0↑0.2 | 52.6↓0.4 | 75.1↑0.1 | 22.1↓0.3 |
| **+ Med-Scout (RL)** | **52.1**↑3.3 | **70.1**↑3.1 | **71.0**↑3.2 | **55.9**↑2.9 | **75.4**↑0.4 | **22.7**↑0.3 |
| Δ | +3.6 | +3.0 | +3.0 | +3.3 | +0.3 | +0.6 |
| *Lingshu-7B* | | | | | | |
| Baseline | 61.2 | 68.9 | 82.8 | 56.3 | 81.4 | 27.4 |
| + Med-Scout (SFT) | 61.3↑0.1 | 69.0↑0.1 | 82.9↑0.1 | 55.9↓0.4 | 81.1↓0.3 | 27.0↓0.4 |
| **+ Med-Scout (RL)** | **64.0**↑2.8 | **71.0**↑2.1 | **83.0**↑0.2 | **57.4**↑1.1 | **81.9**↑0.5 | **28.0**↑0.6 |
| Δ | +2.7 | +2.0 | +0.1 | +1.5 | +0.8 | +1.0 |

## C.8. Case Study

To qualitatively demonstrate that our framework effectively compels models to ground their reasoning in intrinsic physical logic, we present two representative case studies in Figure 13 and Figure 14.

As illustrated in the examples, the baseline Qwen3-VL-8B-Instruct model demonstrates a classic symptom of geometric blindness. In both the chest radiograph (Figure 13, top) and the abdominal CT (Figure 14, bottom), the baseline model successfully identifies the high-level semantic features of the pathologies, generating accurate clinical terms such as "heterogeneously enhancing mass" and "cavitary pulmonary opacity." However, it completely fails in spatial grounding, mislocalizing the lesions to the contralateral side (hallucinating "right kidney" instead of the left, and "left upper hemithorax" instead of the right). This spatial inversion indicates that the baseline model relies heavily on statistical language priors rather than true visual-spatial perception. Conversely, the model aligned with Med-Scout completely rectifies these topological errors. It provides precise, geometrically faithful localizations. This stark contrast compellingly confirms that Med-Scout effectively cures spatial hallucinations, successfully aligning the model's semantic text generation with the objective, intrinsic physical logic of the medical images.

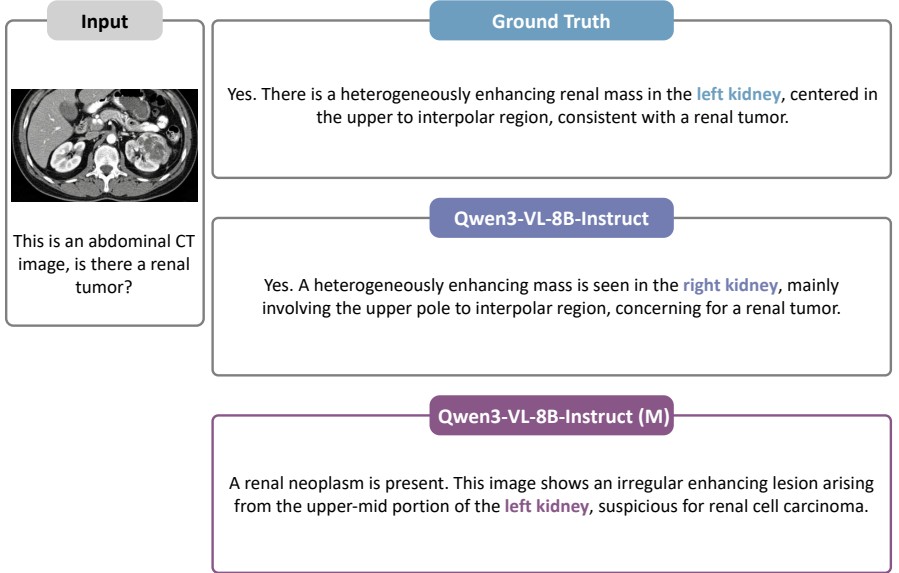

**Input**

Please describe the findings of this image.

**Ground Truth**

Single frontal chest radiograph obtained. There is a large cavitary air-space opacity in the **right upper to mid lung, centered in the right upper lobe/perihilar region**, with surrounding patchy consolidation. Left lung is clear. No pleural effusion or pneumothorax. Cardiomediastinal silhouette is within normal size limits.

**Qwen3-VL-8B-Instruct**

A frontal chest X-ray reveals a cavitary pulmonary opacity in the **left upper hemithorax**. Patchy surrounding air-space density is also noted. The right lung is otherwise clear. No pleural fluid collection or pneumothorax is seen. Cardiomediastinal contours are within normal limits.

**Qwen3-VL-8B-Instruct (M)**

The chest radiograph demonstrates a thick-walled cavitary parenchymal opacity projecting over the **right upper lung, extending into the right mid lung and centered near the right suprahilar/perihilar region**.

*Figure 13.* Case Study 1. While the baseline model correctly identifies the pathology semantics, it suffers from severe geometric blindness, mislocalizing the findings to the opposite side of the body. Med-Scout corrects these spatial hallucinations, proving its capability to ground reasoning in intrinsic physical logic.

**Input**

This is an abdominal CT image, is there a renal tumor?

**Ground Truth**

Yes. There is a heterogeneously enhancing renal mass in the **left kidney**, centered in the upper to interpolar region, consistent with a renal tumor.

**Qwen3-VL-8B-Instruct**

Yes. A heterogeneously enhancing mass is seen in the **right kidney**, mainly involving the upper pole to interpolar region, concerning for a renal tumor.

**Qwen3-VL-8B-Instruct (M)**

A renal neoplasm is present. This image shows an irregular enhancing lesion arising from the upper-mid portion of the **left kidney**, suspicious for renal cell carcinoma.

*Figure 14.* Case Study 2. While the baseline model correctly identifies the pathology semantics, it suffers from severe geometric blindness, mislocalizing the findings to the opposite side of the body. Med-Scout corrects these spatial hallucinations, proving its capability to ground reasoning in intrinsic physical logic.

## D. Theoretical Analysis: Geometric Manifold Alignment

To rigorously justify that Med-Scout cures geometric blindness rather than merely overfitting to the proxy task templates, we analyze the learning process through the lens of Manifold Learning and Energy-Based Models (EBMs).

**Formulation: Geometric Blindness as Manifold Deviation.** Let the space of valid medical visual-text pairs lie on a high-dimensional manifold $\mathcal{M}_{geo} \subset \mathcal{X} \times \mathcal{Y}$. A pair $(x, y)$ is geometrically valid if and only if it satisfies a set of intrinsic physical constraints $\mathcal{C}$ (e.g., anatomical topology, scale consistency):

$$\mathcal{M}_{\text{geo}} = \{(x, y) \mid \mathcal{C}_{\text{scale}}(x, y) \wedge \mathcal{C}_{\text{topo}}(x, y) \wedge \mathcal{C}_{\text{anom}}(x, y) = \text{True}\} \tag{8}$$

*Geometric Blindness* occurs when an MLLM learns a strictly semantic distribution $P_\theta(y|x)$ that covers a broader, "halluci-nated" manifold $\mathcal{M}_{\text{halluc}} \supset \mathcal{M}_{\text{geo}}$. Within $\mathcal{M}_{\text{halluc}}$, plausible but geometrically impossible descriptions (e.g., "liver on the left") are assigned high probability (low energy), indistinguishable from factual descriptions.

**Proxy Tasks as Manifold Constraints.** Our three proxy tasks function not as simple Q&A pairs, but as constraint operators that explicitly penalize deviations from $\mathcal{M}_{\text{geo}}$. By optimizing the dense geometric reward, we are essentially minimizing the energy of the model distribution specifically on the manifold $\mathcal{M}_{\text{geo}}$. The objective of Med-Scout is to reshape the energy landscape $E(x, y) = -\log P_\theta(y|x)$ such that:

$$E(x, y_{\text{halluc}}) \gg E(x, y_{\text{truth}}), \quad \forall y_{\text{halluc}} \notin \mathcal{M}_{\text{geo}} \tag{9}$$

Specifically, the tasks enforce multi-scale correspondence ($\mathcal{T}_{\text{scale}}$), topological integrity ($\mathcal{T}_{\text{topo}}$), and fine-grained structural consistency ($\mathcal{T}_{\text{anom}}$).

**Proof of True Grounding via Energy Landscapes.** If the model were merely overfitting to the templates of the proxy tasks (e.g., memorizing specific grid indices), the energy landscape reshifting would be confined strictly to the subspace of those templates. However, our empirical analysis on natural language reports (Figure 6) proves this is not the case.

In Figure 6, we utilized a probe dataset of factual vs. counterfactual reports derived from MIMIC-CXR.

- **Baseline (Blindness):** The overlapping energy distributions indicate that the baseline model treats the true manifold and the hallucinated space $\mathcal{M}_{\text{halluc}}$ as equiprobable.

- **Med-Scout (Aligned):** The emergence of a distinct energy barrier on this natural language task serves as a theoretical certificate. It demonstrates that the constraints learned from the proxy tasks have successfully propagated to the general probability density function of the model.

This confirms that Med-Scout has successfully internalized the intrinsic boundaries of the medical geometric manifold $\mathcal{M}_{\text{geo}}$, rather than merely minimizing loss on a specific set of training artifacts.

## E. Perspectives

While Med-Scout demonstrates significant improvements in curing geometric blindness, this study also opens several promising directions for future research. Beyond simply scaling models or expanding datasets, we believe the broader value of Med-Scout lies in establishing a clinically inspired paradigm for teaching MLLMs to perceive medical images through verifiable visual logic.

**Validation on Larger Model Scales.** Our experiments were primarily conducted on MLLMs with parameters ranging from 3B to 8B, including Qwen3-VL-4B/8B, Lingshu-7B, and HuatuoGPT-Vision-7B. This choice was mainly driven by computational resource constraints rather than any inherent limitation of the proposed framework. Med-Scout is designed to be model-size agnostic, since its core supervision comes from geometric constraints derived from the image itself rather than from model-specific architectures. Therefore, the proposed geometric alignment principles can be naturally applied to larger-scale foundation models, such as 70B-level or even stronger MLLMs.

Scaling Med-Scout to larger models may bring two complementary benefits. On the one hand, larger models usually possess stronger semantic priors and broader medical knowledge, which may help them integrate geometric evidence with clinical concepts more effectively. On the other hand, Med-Scout can regularize such models by forcing their language generation to respect objective visual facts, thereby reducing the risk that stronger linguistic ability leads to more fluent but geometrically incorrect hallucinations. This suggests an important future direction: combining the rich knowledge of large foundation models with explicit geometry-aware post-training to build medical MLLMs that are not only knowledgeable, but also visually faithful.

**Learning from Clinical Reading Logic.** A central insight of Med-Scout is that geometric perception should not be treated as an isolated low-level visual skill. In clinical practice, physicians rarely read medical images by passively recognizing isolated visual patterns. Instead, they follow a systematic reasoning process: they first build a global anatomical impression, then zoom into suspicious regions, compare local structures with surrounding tissues or adjacent slices, and finally verify whether the observed findings are consistent with anatomical topology and clinical priors. This structured reading behavior is an important source of inspiration for Med-Scout.

The three proxy tasks in Med-Scout can be viewed as computational abstractions of these clinical habits. Hierarchical scale localization mimics the clinician's global-to-local reading process, where a finding must be anchored in both the whole image and a local region. Topological jigsaw reconstruction reflects the use of anatomical layout and spatial continuity to infer whether structures are placed correctly. Anomaly consistency detection resembles comparative scrutiny, where physicians identify subtle discontinuities by comparing a suspicious region with nearby tissues, adjacent slices, or visually similar references. In this sense, Med-Scout does not merely construct artificial pretext tasks; it distills clinical visual reasoning into objective and verifiable learning signals.

This perspective points to a broader research direction: future medical MLLMs should learn not only from diagnostic labels or textual reports, but also from the cognitive procedures used by clinicians during image interpretation. Many clinical reasoning patterns can potentially be transformed into proxy tasks, such as tracing anatomical continuity across slices, checking left-right symmetry, comparing temporal changes across follow-up scans, verifying lesion-organ relationships, or distinguishing true abnormalities from imaging artifacts. By converting these human reading strategies into scalable training objectives, future work can move beyond imitation of final clinical conclusions and instead teach MLLMs how to observe, compare, verify, and reason like medical experts.

**From Clinical Insight to MLLM Optimization.** Another important future direction is to systematically bridge clinical insight and MLLM optimization. Clinical reasoning is often implicit, experience-driven, and difficult to annotate at scale. However, many of its underlying principles are grounded in visual regularities that already exist within unlabeled medical images. Med-Scout shows that these regularities can be mined through proxy tasks and optimized with dense rewards. This provides a practical pathway for transforming expert reading logic into machine-learnable supervision without requiring exhaustive manual annotation.

Future work can further expand this idea by designing a richer library of clinically grounded proxy tasks. For example, temporal comparison tasks can be derived from longitudinal scans to teach progression reasoning; cross-slice consistency tasks can be constructed from volumetric CT and MRI to enhance 3D anatomical understanding; symmetry-based tasks can encourage models to detect unilateral abnormalities; and report-image consistency tasks can help align textual descriptions with precise visual evidence. These tasks would allow MLLMs to internalize the intermediate steps of clinical perception, rather than merely optimizing for final answer accuracy. Such a direction may help close the gap between language-based medical knowledge and the perceptual discipline required for reliable clinical decision support.

**Scope of Medical Modalities.** Our current training data and Med-Scout-Bench focus on CT, MRI, and X-ray images. We selected these three modalities because they are representative forms of structural medical imaging and contain strong geometric constraints, including anatomical scale, spatial topology, and structural continuity. However, the core philosophy of Med-Scout, namely mining intrinsic visual logic from unlabeled data through clinically grounded proxy tasks, is not limited to these modalities.

The same principle can be extended to a broader spectrum of medical imaging. In pathology, models may learn to reason across different magnifications in whole-slide images, reflecting how pathologists switch between low-power tissue organization and high-power cellular details. In ultrasound, proxy tasks may focus on structural continuity, boundary consistency, and view-dependent anatomical changes. In dermoscopy, models may benefit from tasks that emphasize lesion symmetry, border irregularity, and local texture consistency. These extensions require modality-specific task designs, but the underlying goal remains the same: to extract reliable visual reasoning signals from the internal structure of medical images.

**Toward Clinically Faithful Medical MLLMs.** Ultimately, Med-Scout suggests that the next stage of medical MLLM development should move from answer imitation toward perception alignment. A clinically reliable model should not only generate plausible medical language, but also ground each conclusion in the physical and geometric truth of the image. This requires models to develop habits similar to clinical image reading: observing at multiple scales, respecting anatomical topology, comparing subtle differences, and verifying visual consistency before producing a conclusion.

By demonstrating that such habits can be approximated through unlabeled data, proxy tasks, and geometry-aware reinforcement learning, Med-Scout provides a scalable step toward this goal. We envision future medical MLLMs as systems that combine three forms of intelligence: the semantic knowledge of foundation models, the perceptual discipline of geometry-aware optimization, and the structured reasoning logic inspired by clinicians. This integration may lead to medical AI systems that are more robust, more interpretable, and more faithful to clinical visual evidence, thereby providing stronger support for trustworthy medical understanding and improving the reliability of downstream clinical applications.

