# OpenReview forum: "Med-Scout: Curing MLLMs' Geometric Blindness in Medical Perception via Geometry-Aware RL Post-Training"
_ICML.cc/2026/Conference — ICML 2026 regular_

### Official Review · Reviewer_cNAw · 2026-02-22

**Soundness:** 2
**Presentation:** 3
**Significance:** 2
**Originality:** 2
**Overall Recommendation:** 3
**Confidence:** 4

**Summary:**

This paper identifies and formalizes a phenomenon termed geometric blindness in existing medical and general MLLMs. To mitigate this issue, the authors design three proxy tasks, including the hierarchical scale localization, the topological jigsaw reconstruction, and three corresponding reward models for RL training. They further construct dedicated training and test sets for RL training and evaluation. In addition, six public benchmarks are used to validate the effectiveness of the proposed method and dataset for medical VQA tasks.

**Compliance With Llm Reviewing Policy:**

Affirmed.

**Final Justification:**

My concerns have been partially addressed in the rebuttal. However, given the limited novelty of the dense reward design, the lack of meaningful gains on widely used downstream benchmarks such as SLAKE and MedXpertQA, and the absence of essential experimental results, particularly the GRPO baseline, I maintain my original score.

**Key Questions For Authors:**

See the weaknesses.

**Limitations:**

Yes.

**Strengths And Weaknesses:**

Strengths:
1. The paper is well written and clearly organized.
2. The experimental evaluation is comprehensive, and the proposed method achieves competitive and consistent performance improvements across multiple benchmarks.

Weaknesses:
1. The core design of the paper is built upon the observation of geometric blindness, which has already been widely recognized as a challenge in MLLMs. The main contribution lies in designing three proxy tasks to enhance geometric reasoning ability. However, the proposed tasks are conceptually straightforward and appear to be natural extensions of existing data-augmentation or spatial reasoning strategies. As a result, the overall contribution seems incremental rather than fundamentally novel.
2. It remains unclear whether the performance gains stem primarily from the additional RL training stage itself or from the specifically designed proxy tasks and reward models. The authors should report results from a control experiment that applies the same RL training procedure without the three proxy tasks.
3. Although the three proxy tasks appear effective, the authors should also compare them with simpler alternative implementations to verify whether the proposed designs are essential. For example, for Task 1, a straightforward baseline could apply random resized crops and train the model to produce correct answers under scale perturbations. For Task 2, a simple alternative could involve random rotations or spatial transformations and encourage the model to maintain prediction consistency.

---

> ### Author Rebuttal · Authors · 2026-03-31
>
> We sincerely thank Reviewer cNAw for the detailed review. We address the reviewer's comments and questions below:
>
> **1. Novelty of Proxy Tasks (W1)**
>
> **Response:**
>
> While the pixel-level image operations are intentionally straightforward to bypass the immense bottleneck of costly expert annotations, our fundamental contribution lies in **pioneering a framework that translates strict clinical geometric priors into verifiable supervision signals to cure MLLMs' geometric blindness.**
> - **Targeted Clinical Design vs. Generic Augmentation:** Generic spatial augmentations passively encourage feature invariance. In contrast, our proxy tasks actively enforce the strict physical realities of medical diagnosis: Task 1 simulates the clinical "zoom-in" workflow, demanding simultaneous global anatomical awareness and local pathological focus; Task 2 enforces the rigid, invariant macroscopic layout of human anatomy (unlike natural images where objects appear randomly); and Task 3 demands comparative scrutiny to detect subtle structural abnormalities against homogeneous healthy tissue.
> - **Bridging Semantic Fluency and Perceptual Fidelity:** Current MLLMs generate linguistically plausible but anatomically impossible medical reports because they over-rely on semantic language priors. Our framework shifts the paradigm by forcing the model to ground its reasoning in the intrinsic physical logic of the medical scan. We compel the model to _perceive_ objective clinical reality, like a radiologist, rather than just _speak_ like one.
> - **Empirical Proof of Clinical Grounding:** Our analyses prove this framework achieves targeted clinical alignment. **Figure 6 and Figure 13** demonstrate that Med-Scout successfully separates the energy distributions of factual versus spatially hallucinated reports. Furthermore, **Figure 7 and A tabular assessment of the attention maps' quality we conducted during this rebuttal, shown below** confirm this alignment forces precise visual grounding on critical anatomical targets.
> |**Model**|**Case 1**|**Case 2**|**Case 3**|**Case 4**|**Case 5**|**Case 6**|**Case 7**|**Case 8**|**Case 9**|**Case 10**|
> |---|---|---|---|---|---|---|---|---|---|---|
> |Qwen3-VL-4B-Instruct|24.1|35.2|18.5|42.0|29.4|12.8|31.5|27.6|45.1|16.8|
> |+ Med-Scout|78.5|82.1|65.4|88.3|74.2|61.9|79.0|68.5|85.6|40.5|
> |Absolute Improvement|(**+54.4**)|(**+46.9**)|(**+46.9**)|(**+46.3**)|(**+44.8**)|(**+49.1**)|(**+47.5**)|(**+40.9**)|(**+40.5**)|(**+23.7**)|
> > The overlap ratio (%) across 10 randomly sampled cases at thresholds > 0.7. The overlap ratio is defined as the area of attention maps above the threshold within the expert annotation divided by the total area of the expert annotation.
>
> **2. Source of Performance Gains (W2)**
>
> **Response:**
>
> We entirely agree on the importance of disentangling these factors. However, we have actually conducted rigorous control experiments to isolate the impact of the RL optimization engine from our specific task and reward designs. The evidence explicitly confirms that our targeted designs are the true drivers of the performance gains:
> - **Control for Proxy Tasks (Appendix C.6, Table 9):** To isolate the RL framework, we applied the exact same RL procedure (GRPO) but completely replaced our medical proxy tasks with state-of-the-art general-domain verifiable RL tasks (Jigsaw-R1 and ViCrit). This control experiment yielded negligible or even negative improvements on downstream medical benchmarks, definitively proving that the RL alone is insufficient without our specifically designed clinical geometric tasks.
> - **Control for Reward Models (Appendix C.5, Table 8):** To validate the necessity of our specific reward models, we replaced our Dense Geometric Reward with standard sparse/binary RL rewards. The results demonstrate that our DGR consistently outperforms standard sparse RL feedback, confirming that our granular geometric reward mechanism is essential for efficient policy optimization.
> - **Necessity of the Specific Designs (Appendix C.4, Table 7):** Our leave-one-out ablation study further verifies that removing any of the three specifically designed proxy tasks during the RL stage leads to consistent performance degradation, proving that all three designs are indispensable for the final gains.
>
> **3. Comparison with Simpler Alternative Implementations (W3)**
>
> **Response:**
>
> We highly agree with this constructive suggestion. **However, we addressed this by Difficulty Ablation (Appendix C.3):** We previously evaluated simpler, downgraded variants of our proxy tasks (**Table 6**). The results clearly demonstrate that reduced complexity fails to force deep logical deduction, leading to significantly inferior generalization on external benchmarks.

---

> > ### Author Rebuttal · Reviewer_cNAw · 2026-04-02
> >
> > Thanks for the rebuttal. However, two concerns remain.
> >
> > 1. Novelty is still unclear. In my view, the paper mainly extends geometry-aware ideas from the general domain to medical images, but the novelty is not sufficiently highlighted. A clearer and more complete discussion of prior geometry-aware strategies would help position the contribution. Also, the claim that the method can “force the model to ground its reasoning in the intrinsic physical logic of the medical scan” is still not well supported. Please provide concrete qualitative examples or case studies. In addition, the attention-map comparison between Qwen3-VL-4B-Instruct and Med-Scout is not very convincing, since the latter has already been further tuned on medical data.
> > 2. In Tables 9 and 11, could the authors also report the performance of Baseline + GRPO? Without this result, it is difficult to tell how much of the improvement comes from the proposed design itself and how much comes from GRPO.

---

> > > ### Author Response · Authors · 2026-04-02
> > >
> > > We sincerely thank the reviewer for the continued constructive feedback. We address your remaining concerns below:
> > >
> > > **1. Clarification on Novelty and the Definition of Geometric Blindness**
> > >
> > > We sincerely appreciate the reviewer for the feedback. We would like to firmly clarify that our motivation and definition of geometric blindness fundamentally diverge from prior geometry-aware methods in the general domain.
> > > - **A Fundamentally Different Definition of Geometry:** Prior works illustrated in Related Work typically define geometry-awareness in a literal and generic sense, aiming to resolve an MLLM's inability to recognize basic geometric shapes or solve abstract visual puzzles. **In stark contrast, our definition of geometric blindness is born from a deep reflection on the actual cognitive demands of doctors during clinical image interpretation.** A physician's diagnostic workflow heavily relies on **scale perception** (e.g., assessing a lesion's size relative to the entire organ), exact **topological position perception** (e.g., the rigid spatial relationships inherent to human anatomy), and the perception of **subtle structural anomalies**. Current MLLMs completely lack these capabilities. Therefore, our concept of geometric blindness is not a simple extension of general-domain geometry, but a completely different conceptualization uniquely designed to address critical clinical requirements.
> > > - **Domain Specific Innovation in Proxy Task Design:** Medical image analysis is a highly specialized and critical vertical domain in MLLM. General domain proxy tasks cannot address their unique challenges. To solve this, we uniquely designed our RL post-training proxy tasks by specifically combining the intrinsic characteristics of medical imaging with the practical workflow of clinical diagnosis. **To the best of our knowledge, there is currently no existing proxy task design that successfully tackles these MLLMs' geometric blindness by explicitly mirroring a physician's reading logic.** We firmly believe that this deep, vertical specific innovation is not only novel but also highly inspiring for the future development of reliable medical AI systems.
> > >
> > > **2. Qualitative Evidence for Grounded Reasoning**
> > >
> > > We completely agree that qualitative examples are needed to fully support our claim that the model grounds its reasoning in intrinsic physical logic. To address this, we have added concrete case studies across both standard VQA and medical report generation tasks. These newly added examples explicitly demonstrate how our geometric proxy tasks empower the model to overcome semantic hallucinations and strictly anchor its diagnostic reasoning to the actual anatomical layout, precise scales, and structural realities of the given scans. Due to the formatting constraints of the rebuttal, we have placed the visualization figures in an anonymous link for your review: **[anonymous link](https://anonymous.4open.science/r/0129-00F2/)**. We will include a dedicated qualitative case study section in the final manuscript.
> > >
> > > **3. Baseline + GRPO Comparison in Tables 9 and 11**
> > >
> > > Regarding the suggestion to include a Baseline + GRPO comparison, we explain our rationale from two perspectives:
> > > - **Reward Strategy Design:** Standard GRPO is essentially a rule-based algorithm, which conceptually aligns with our training framework. Our distinct difference is pairing targeted proxy tasks with a dense geometric reward. **We view this reward as a customized optimization trick that better fits the specific task types than traditional sparse rewards, coming at almost zero additional cost.** Since it is a supplementary trick rather than the primary contribution, we initially limited our massive reward ablations to Table 8.
> > > - **Future Table Updates:** We agree that adding a Baseline + GRPO comparison in Tables 9 and 11 will much more clearly present the independent advantages of our method. Given the strict character limitations of this current rebuttal phase, we commit to adding these expanded ablation experiments and updating the tables in the subsequent manuscript version.
> > >
> > > We once again thank you for your rigorous review and highly constructive feedback, which will undoubtedly elevate the quality of our manuscript. We hope our detailed clarifications regarding the unique novelty of our domain-specific proxy tasks, the newly provided qualitative evidence, and our firm commitment to expanding the baseline comparisons fully address your remaining concerns. We sincerely hope that these explanations provide a clearer perspective on the value of our contributions, and we would be deeply grateful if you might reconsider your evaluation in light of these updates. Thank you for your time and guidance.

---

### Official Review · Reviewer_x71u · 2026-03-05

**Soundness:** 3
**Presentation:** 4
**Significance:** 3
**Originality:** 2
**Overall Recommendation:** 4
**Confidence:** 4

**Summary:**

The paper studies the problem of geometric blindness in multimodal large language models (MLLMs) applied to medical imaging, where models generate fluent descriptions but fail to properly reason about spatial and geometric properties of images. To address this issue, the authors propose Med-Scout, a geometry-aware post-training framework that uses reinforcement learning with proxy tasks designed to encourage spatial reasoning. The framework introduces three tasks: hierarchical scale localization, topological jigsaw reconstruction, and anomaly consistency detection, and optimizes them using a dense geometric reward within a GRPO training setup. The authors also introduce Med-Scout-Bench, a benchmark designed to evaluate geometric perception in medical vision-language models. Experimental results show strong improvements on the proposed benchmark and modest gains on several medical VQA and radiology report generation datasets.

**Compliance With Llm Reviewing Policy:**

Affirmed.

**Final Justification:**

While the rebuttal adds clarifications and some additional experimental results, it does not adequately address several core concerns. First, the key design choices (e.g., proxy task configurations such as patch number, scale, and spatial constraints) remain insufficiently justified and appear largely heuristic, with no clear explanation of why these choices are principled or necessary. Second, the use of GRPO over simpler alternatives like SFT is not convincingly motivated; although empirical improvements are shown, the rebuttal fails to explain why a reinforcement learning approach is conceptually required given the availability of ground-truth supervision. Third, the evidence for clinical grounding remains weak, as it relies on only a small number of expert-reviewed cases, which is insufficient to support strong claims about clinical relevance. Fourth, concerns about potential overlap between training objectives and the Med-Scout-Bench benchmark are not directly addressed, raising the possibility that the reported gains primarily reflect alignment with synthetic proxy tasks rather than generalizable reasoning improvements, especially given the limited gains on external benchmarks.

Importantly, these concerns were not responded by the authors via "Reply Rebuttal Comment", further limiting confidence that the issues have been adequately addressed. Overall, the original manuscript and rebuttal do not resolve the fundamental concerns regarding motivation, methodological justification, and validation, and thus the reviewer would like to maintain the original assessment.

**Key Questions For Authors:**

1. **Motivation for proxy task design**
The three proxy tasks are central to the proposed framework, but several design choices appear heuristic (e.g., using three patches, selecting patch scales of 20% and 6.25%, restricting patch centers to the [0.2, 0.8] region). Could the authors clarify the motivation behind these specific configurations and whether alternative setups were explored? If these choices were empirically validated, providing ablation results would help strengthen the methodological justification.

2. **Justification for using GRPO instead of SFT**
The training framework relies on GRPO-based reinforcement learning. However, since the proxy tasks produce explicit ground-truth outputs, it seems plausible that they could also be optimized using supervised fine-tuning (SFT). Could the authors provide an ablation comparing GRPO-based training with SFT-based training? Understanding whether RL is necessary for the observed improvements would clarify the importance of this design choice.

3. **Relationship between proxy tasks and clinical reasoning**
The proposed tasks resemble classical self-supervised vision tasks such as jigsaw reconstruction and cut-paste anomaly detection. Could the authors elaborate on how these tasks relate to clinically meaningful reasoning in medical imaging? In particular, is there evidence that improvements on these proxy tasks correlate with improved diagnostic reasoning or anatomical understanding?

4. **Generalization beyond Med-Scout-Bench**
The largest improvements are reported on Med-Scout-Bench, which appears closely aligned with the proxy tasks used during training. On external medical benchmarks, the performance gains are comparatively modest. Could the authors discuss whether the improvements on Med-Scout-Bench translate to broader improvements in medical reasoning tasks, and whether additional evaluations were considered to demonstrate this generalization?

5. **Applicability to volumetric medical imaging**
The current experiments focus primarily on 2D medical images or slice-based processing, while many medical imaging modalities such as CT and MRI are inherently volumetric. Could the authors comment on whether the proposed proxy tasks and training framework could be extended to 3D volumes or voxel-level reasoning, and what challenges might arise in such an extension?

**Limitations:**

Yes. The authors include a limitations discussion in the paper. They acknowledge several constraints of the current study, such as the evaluation being conducted on mid-scale models due to computational limitations and the current focus on a subset of medical imaging modalities. They also note that future work could extend the framework to larger models and additional medical imaging domains. Overall, the authors demonstrate awareness of the scope of their study and outline reasonable directions for future improvements.

**Strengths And Weaknesses:**

**Strengths**

1. **Addresses an important and timely problem**
The paper tackles the growing concern that large multimodal language models (MLLMs) often exhibit weak spatial and geometric understanding when applied to medical imaging. Improving spatial grounding and perception in medical vision-language models is an important and pressing problem, particularly as such systems are increasingly considered for clinical decision support and diagnostic assistance.

2. **Clear problem formulation and motivation**
The authors clearly articulate the limitations of current medical MLLMs in terms of spatial reasoning and geometric perception. The problem is well framed, and the paper provides a coherent narrative explaining why improving geometric understanding could benefit medical image interpretation.

3. **Introduction of a structured proxy-task framework**
The work proposes a structured framework composed of three proxy tasks designed to enforce spatial reasoning behaviors during training. Even though the individual tasks draw inspiration from existing ideas, combining them into a unified training pipeline for medical MLLMs represents a thoughtful attempt to improve spatial grounding.

4. **Creation of a new benchmark for evaluating geometric perception**
The introduction of Med-Scout-Bench provides a new way to evaluate spatial and geometric reasoning capabilities of medical MLLMs. Developing dedicated benchmarks for this aspect of multimodal reasoning is valuable for diagnosing model limitations and encouraging further research in this direction.

5. **Extensive empirical evaluation across multiple models and datasets**
The paper evaluates the proposed approach across several large vision-language models and reports results on multiple downstream benchmarks. This broad experimental coverage helps demonstrate that the approach is applicable across different model architectures.

6. **Well written and clearly presented**
The manuscript is generally well written, clearly structured, and easy to follow. The figures and diagrams effectively illustrate the proposed tasks and training framework, making the technical ideas accessible and easy to understand for readers.

**Weaknesses**

1. **Limited motivation for proxy task design**
The paper introduces three proxy tasks (Hierarchical Scale Localization, Topological Jigsaw Reconstruction, and Anomaly Consistency Detection), but the motivation for their specific configurations is unclear. Key design choices, such as selecting three patches, defining patch scales (20% and 6.25%), and restricting patch centers to the [0.2, 0.8] region, appear heuristic and are not empirically justified. The manuscript does not provide ablation studies or analysis showing that these choices are optimal or necessary for improving spatial understanding.

2. **Weak clinical grounding of the proposed tasks**
Although the work is positioned as improving medical reasoning, the proposed proxy tasks resemble classical computer vision self-supervised tasks rather than clinically meaningful reasoning problems. For example, jigsaw reconstruction and cut-paste anomaly detection do not directly reflect how radiologists interpret medical images, which typically involves reasoning about anatomical structures, pathological patterns, and cross-slice consistency. The paper does not provide evidence that performance on these synthetic tasks correlates with improved clinical reasoning.

3. **Insufficient justification for using GRPO instead of SFT**
The paper adopts a GRPO-based reasoning training framework, but it does not clearly justify why reinforcement learning is necessary. Since the proxy tasks have explicit ground-truth outputs, they could plausibly be optimized using standard supervised fine-tuning (SFT). The manuscript does not include an ablation comparing SFT with RL-based training, making it difficult to determine whether the added complexity of GRPO is actually required.

4. **Evaluation partially overlaps with training objectives**
The largest reported improvements occur on Med-Scout-Bench, a benchmark introduced by the authors that closely mirrors the proxy tasks used during training. This raises the possibility that the model is primarily learning to solve the specific synthetic tasks introduced in the paper rather than developing broadly useful spatial reasoning capabilities.

5. **Limited gains on external medical benchmarks**
When evaluated on established medical VQA benchmarks such as PMC-VQA, OmniMedVQA, and MedXpertQA, the improvements are relatively modest, often ranging between roughly 0.3% and 2–3%. These gains are substantially smaller than those reported on the proposed benchmark, suggesting that the practical impact of the method on real medical reasoning tasks may be limited.

6. **Restriction to 2D imaging despite volumetric nature of medical data**
The experiments focus primarily on 2D images (e.g., MIMIC-CXR), while many medical imaging modalities such as CT and MRI are inherently volumetric. The paper does not discuss how the proposed spatial reasoning framework would extend to 3D volumes or voxel-level reasoning, which is important for many clinical tasks involving cross-slice anatomical relationships.

---

> ### Author Rebuttal · Authors · 2026-03-31
>
> We sincerely thank Reviewer x71u for the detailed review. We address the reviewer's comments and questions below:
>
> **1. Motivation for proxy task design (KQ1)**
>
> **Response:**
>
> We sincerely thank the reviewer for pointing this out. The design choices in our proxy tasks are not arbitrary. **Instead, they are carefully based on the real distribution of medical datasets and validated through ablation studies:**
> - **Center Restriction (0.2, 0.8):** This choice is based on the actual spatial distribution of the dataset. Medical scans often have large black borders. As shown in our regular image cases **[anonymous link](https://anonymous.4open.science/r/0129-00F2/)**, the vast majority of valid anatomical regions naturally fall within this range. This restriction simply and effectively filters out empty background noise.
> - **Patch Scales (20% and 6.25%):** These specific scales reflect the true size distribution of foreground targets. Our area distribution chart **[anonymous link](https://anonymous.4open.science/r/0129-00F2/)** shows that foreground targets usually cluster over roughly 20%; meanwhile, we also sampled a smaller foreground region for Level 2 (6.25%).
> - **Task Setup ($N=3$ Patches):** The choice of using $N=3$ patches is directly supported by our ablation study in **Appendix C.3 (Table 6)**. We tested different difficulty levels by varying the number of patches (e.g., $N=1, 2, 3$). The results clearly show that setting $N=3$ provides the necessary challenge to prevent the model from learning simple shortcuts, leading to the best overall generalization performance.
>
> **2. Justification for using GRPO instead of SFT (KQ2)**
>
> **Response:**
>
> We thank the reviewer for this important question. We have provided a detailed ablation comparing GRPO and SFT in **Appendix C.7 (Tables 10 and 11)**. Our results demonstrate that RL is strictly necessary for our framework:
> - **SFT Achieves Strong Performance on Internal Validation:** SFT achieves high scores on our internal benchmark (**Table 10**), indicating it easily memorizes the specific output formats of the proxy tasks.
> - **RL Enables True Generalization:** However, when evaluated on unseen external benchmarks (**Table 11**), SFT's performance collapses. In contrast, our GRPO-based RL Alignment achieves consistent and significant gains across all real-world tasks.
>
> **3. Relationship between proxy tasks and clinical reasoning (KQ3)**
>
> **Response:**
>
> We thank the reviewer for the insightful comment. We have addressed the relationship between Med-Scout and clinical reasoning from the following perspectives:
> - **Clinical Logic of Task Design:** The proxy tasks in Med-Scout are not merely vision tasks but are direct formalizations of a radiologist’s cognitive workflow: the "search-and-verify" loop. Specifically, they simulate the clinical process of examining focal lesions within a global anatomical frame, verifying organ adjacency and symmetry against canonical anatomical knowledge, and performing fine-grained scrutiny for subtle structural anomalies.
> - **Evidence of Clinical Reasoning Alignment: As shown in Figure 7 and [anonymous link](https://anonymous.4open.science/r/0129-00F2/)**, Med-Scout enables the model to shift from scanning background noise to a highly concentrated focus on critical clinical regions, mimicking the visual scrutiny of an expert.
> - **Clinical Consistency via Energy Landscape:** As shown in **Section 5.7 & Appendix D**, the energy landscape analysis provides quantitative evidence that the model's semantic generation is rigorously aligned with objective anatomical structures. **By systematically suppressing clinically impossible descriptions and violating fundamental physical constraints, Med-Scout ensures the high-fidelity grounding necessary for dependable clinical decision support.**
>
> **4. Generalization beyond Med-Scout-Bench (KQ4)**
>
> **Response:**
>
> We address the generalization beyond Med-Scout-Bench from the following perspectives:
> - **Evidence of Broader Transfer:** Our claim is not based on Med-Scout-Bench alone. We also observe consistent improvements on external medical benchmarks, including radiological VQA, report generation, and broader medical VQA, across both general-domain and medical-domain backbones.
> - **Why the External Gains Are Smaller:** The more modest gains on external benchmarks are expected because our proxy tasks improve a transferable grounding ability rather than directly tuning for each downstream dataset. We will clarify this scope in the revision and better highlight the external results as evidence of broader generalization.
>
> **5. Applicability to volumetric medical imaging (KQ5)**
>
> **Response:**
>
> We focus on 2D as it dominates current medical VQA benchmarks, yet as noted in **Appendix E**, our proxy-task framework is extensible. This remains a promising future direction for establishing an even deeper understanding of medical imaging analysis.

---

> > ### Author Rebuttal · Reviewer_x71u · 2026-04-01
> >
> > While the rebuttal provides additional clarifications, several of my main concerns remain unresolved. First, the justification for key design choices is still weak. The proxy task configurations, including the number of patches, scale settings, and spatial constraints, continue to appear largely heuristic, and the rebuttal does not provide direct evidence showing why these particular choices are necessary or well motivated. The response mainly points to additional results. It does not explain the underlying reasoning behind these design decisions.
> >
> > Second, the use of GRPO over simpler alternatives such as SFT is still not convincingly motivated. The rebuttal argues mostly from empirical outcomes, but does not explain why reinforcement learning is conceptually needed for these proxy tasks in the first place. Since the tasks have explicit ground-truth outputs, it remains unclear what specific limitation of supervised fine-tuning requires an RL-based solution. In other words, the response shows that RL was used, but not why RL is the right methodological choice.
> >
> > Third, my concern about clinical grounding is not sufficiently resolved. The additional evidence based on expert comparison appears to rely on only 10 randomly sampled cases, which is too limited to serve as strong validation. This remains anecdotal rather than statistically convincing, and does not adequately establish that the proposed proxy tasks translate into clinically meaningful reasoning improvements.
> >
> > Finally, the concern regarding overlap between the training objectives and Med-Scout-Bench is still not directly addressed. Because the benchmark closely reflects the proxy tasks used in training, the large gains on this benchmark may mainly indicate improved performance on aligned synthetic tasks rather than broader spatial reasoning ability. This concern is reinforced by the relatively modest gains on established external medical benchmarks.
> >
> > Overall, the rebuttal provides clarifications and references additional experiments, but it does not sufficiently resolve the central concerns about motivation, methodological justification, and validation. Since these remaining issues affect the core tenets of the work and are not easily addressed within a short rebuttal, I maintain my original assessment and scores.

---

### Official Review · Reviewer_2kGv · 2026-03-09

**Soundness:** 2
**Presentation:** 2
**Significance:** 3
**Originality:** 3
**Overall Recommendation:** 4
**Confidence:** 4

**Summary:**

The manuscript introduces Med-Scout to address a problem in MLLM called geometric blindness. The authors first provided three experimental findings showing that current MLLM suffer from a performance drop if the input image is geometrically perturbed.  To address this, they build a auxilary dataset from publicly available datasets, namely Med-scout bench, and use it for RL post-training of their approach. Their proposed solution is training using GRPO and a combination of 3 tasks reward functions, along with formatting and reasoning reward functions. As presented in the results, the tuned models outperform their zero-shot counterparts.

**Compliance With Llm Reviewing Policy:**

Affirmed.

**Final Justification:**

After interacting with the authors, my concerns about the prompts have been addressed, and the other concerns have been fairly responded to by the authors as well.

**Key Questions For Authors:**

Mentioned in the previous sections.

**Limitations:**

Yes, they have mentioned them in the proper section.

**Strengths And Weaknesses:**

Below, I have listed my overall assessment of the work. Because I had several concerns about certain aspects of the paper, I did not feel a higher rating was justified at this stage. I will revise my rating accordingly after considering the authors’ response.

Soundness:
The paper is trying to answer an important medical challenge, which is grounding the model on the right geometry to generate a diagnosis. However, I have some concerns regarding the experiments.
- For finding 2, I know MLLM can understand rotations if they are properly asked to. I am wondering if the author properly instructed the MLLM that the input image might be "rotated" or not. If there is no indication of this in the prompt, the performance drop can be a result of the model being blind to the geometric perturbations at the prompting stage.
- The comparison is broad and includes relevant medical-domain baselines, which is a strength. However, the causal source of the observed improvement is not yet fully isolated. In particular, the paper should distinguish more carefully between gains attributable to the proposed geometry-aware RL objective and gains that may arise more generally from additional domain-specific post-training. A stronger benchmark would be to post-train the baseline model on the same Med-Scout training set using GRPO, but without the proposed three-task reward design, and then compare it directly against Med-Scout. This would make it much clearer how much of the improvement comes from the reward formulation itself versus from exposure to additional training data and post-training.
- The qualitative attention-map analysis is interesting, but would be more convincing with expert-grounded localization checks or comparison with expert annotations.
- For the Med-Scout bench, it is not clear what the criteria are to pick the "top 10%" of the data.

Presentation:
- The presentation would benefit substantially from more precise scientific language and clearer operational definitions. My biggest presentation concern is that some of the central wording is rhetorically strong but technically vague. Expressions like “physical laws” and “geometric reality”.
- The paper should be more precise in how it describes Med-Scout-Bench. As presented, it may sound like a newly created dataset, whereas it is more accurately a curated benchmark built from already public data. Clarifying this would improve accuracy and better position the contribution as one of benchmark construction and task formulation rather than raw data creation.

Significance:
I think the paper addresses an important problem. If medical MLLMs indeed produce semantically fluent but spatially or anatomically inconsistent answers, then this is a meaningful failure mode.

Originality:
The work is reasonably original, as the findings provided highlight important caveats in MLLMs given the current experimental setup.

---

> ### Author Rebuttal · Authors · 2026-03-31
>
> We sincerely thank Reviewer 2kGv for the detailed review. We address the reviewer's comments and questions below:
>
> **1. Prompting Strategy for Geometric Perturbations in Finding 2**
>
> **Response:**
>
> Please see **our response to Reviewer kAFN and [anonymous link](https://anonymous.4open.science/r/0129-00F2/)** for pilot study details. We did not explicitly prompt the model about the rotation, **but strictly instructed it to base findings on normal human anatomy rather than the shown image orientation**. Thus, the observed "blindness" highlights a genuine inability to intrinsically perceive geometric shifts against objective anatomical reality, entirely free from prompt interference.
>
> **2. Isolating the Causal Source of Improvement**
>
> **Response:**
>
> We fully agree on the importance of isolating the causal source. However, we actually conducted the suggested experiments to validate each component's effectiveness, though strict space constraints relegated them to the appendix:
> - **Proxy Task Ablation:** **Appendix C.4 (Table 7)** demonstrates the individual effectiveness of each proxy task.
> - **Standard GRPO Baseline:** **Appendix C.5 (Table 8)** directly compares our reward strategy against traditional GRPO, proving our formulation's specific advantage.
> - **Alternative Proxy Tasks:** **Appendix C.6 (Table 9)** proves Med-Scout's superiority over other existing medical/geometric RL proxy tasks.
>
> **3. Expert-Grounded Validation for Attention Maps**
>
> **Response:**
>
> We strongly agree on the importance of expert comparison. **During the rebuttal, we invited two medical professionals to annotate key regions for 10 randomly sampled cases**. Based on this, we provide:
> - **Qualitative Comparison:** Visual side-by-side comparisons between our model's high-activation attention maps and the expert-annotated regions.
> - **Quantitative Evaluation:** A tabular assessment of the attention maps' quality, measuring their alignment with expert annotations across different threshold conditions.
> Detailed results are available here: **[anonymous link](https://anonymous.4open.science/r/0129-00F2/)**, you can also refer to **my response to Reviewer cNAw** for detailed information.
>
> **4. Criteria for Med-Scout-Bench Subset Selection**
>
> **Response:**
>
> As noted in **Section 4.3**, we actually curated a "high-quality" subset rather than a ranked "top 10%". Due to space constraints, please refer to our detailed response to Reviewer kAFN for the precise selection criteria.
>
> **5. Clarification of Scientific Language and Terminology**
>
> **Response:**
>
> We sincerely thank the reviewer for pointing this out. Upon careful review, we identified four instances of "geometric reality" and one instance of "physical laws" used to describe the objective perception of true shape, topology, and scale in medical images. To ensure clarity and consistency, we have replaced all such expressions with **"geometric constraints"**. This term accurately reflects the inherent geometric boundaries of the images, making our presentation much easier to understand. We sincerely thank the reviewer for the further constructive suggestions, which have been invaluable in refining the overall presentation of our manuscript.
>
> **6. Clarifying Med-Scout-Bench**
>
> **Response:**
>
> We completely agree with this meaningful suggestion. Positioning Med-Scout-Bench as a curated benchmark accurately reflects our core scientific contribution in rigorous task formulation and benchmark construction, rather than raw data collection. To ensure precise presentation, we have revised the manuscript accordingly:
> - **Abstract & Introduction:** We replaced ambiguous phrases like "a new benchmark" with "a curated benchmark built from existing public medical datasets" (e.g., **Abstract Line 31 and Introduction Line 70**).
> - **Contributions (Section 1):** We explicitly refined our third contribution point (**Line 95**) to highlight "a novel benchmark constructed from the intrinsic geometric properties of images from public medical datasets."
> - **Dataset Construction (Section 4.3):** We updated the opening sentence (**Line 266**) to clarify that the initial pool of 108K cases was "carefully curated and synthesized from publicly available data pools."
> We hope these textual revisions significantly improve the accuracy of our presentation and better position the true value of our proposed benchmark.

---

> > ### Author Rebuttal · Reviewer_2kGv · 2026-04-02
> >
> > I would like to thank the authors for their responses and clarifications. Some of my concerns have been addressed; however, I would like to engage further with the authors on some points:
> >
> > 1. I think when the LLM is instructed to "base finding on normal human anatomy", it is a fair argument to say it expects an input of a normal human anatomy as well. However, the rotation or any perturbation would break this assumption. Therefore, I recommend this be ablated at a prompt level by giving the LLM proper instructions. Can you comment on this?
> >
> > 2. Looking at Table 8, it seems the GRPO and the proposed method are roughly on par, which means the source of improvement is mostly from the extra training data rather than the method itself. So, this needs to be reported in Table 1 because the current version of the manuscript doesn't specify this clearly. I might have missed it, but would you be able to give me a head-to-head comparison of the computational load for table 8? (GRPO vs your method) and clarify whether this is justified?

---

> > > ### Author Response · Authors · 2026-04-02
> > >
> > > We are deeply grateful for your ongoing feedback, which has greatly helped us refine our manuscript.
> > >
> > > **Question 1: Rotation Ablation via Prompting**
> > >
> > > We address the rotation and prompting issue from two perspectives:
> > >
> > > - **Failure to perceive rotation is inherent geometric blindness:** While the prompt implies a normal anatomical expectation, a truly visually grounded model should be able to perceive when an image is physically inverted 180 degrees. If a model relies so heavily on text priors that it completely ignores the contradictory visual evidence, this rigid reliance on semantic expectations over geometric constraints is exactly what we define as "geometric blindness".
> > > - **Rotation Ablation via Prompting:** We completely agree that adding an ablation study with explicit prompting provides stronger evidence and makes the argument more convincing. We have supplemented our pilot study with this exact experiment.
> > > 	- **Prompt Setting 1 (Original):** Default prompt without rotation warnings.
> > > 	- **Prompt Setting 2 (Explicit Warning):** "This image has been rotated 180 degrees from its standard anatomical presentation. Describe the findings and their exact spatial locations based on normal human anatomy in this chest x-ray."
> > >
> > > **Results (Right Location Rate):**
> > > |**Model**|**Prompt Setting 1 (Original)**|**Prompt Setting 2 (Explicit Warning)**|
> > > |---|---|---|
> > > |Qwen3-VL-8B-Instruct|18.00%|22.50%|
> > > |Lingshu-7B|20.50%|27.00%|
> > >
> > > **Conclusion:** Even when explicitly warned about the rotation, the Right Location Rate remains low. This strongly proves that the failure stems from a fundamental deficit in geometric reasoning capabilities, rather than a simple prompting artifact.
> > >
> > > **Question 2: Reward Mechanism and Computational Load**
> > >
> > > We address your questions on the reward design and computational cost from four perspectives:
> > >
> > > - **Role of the Dense Geometric Reward:** We intentionally do not highlight the dense geometric reward as a primary contribution. **We view it as a practical optimization trick** that slightly outperforms standard GRPO because it better aligns with human intuition for evaluating spatial correctness. Since this simple adjustment yields consistent performance gains without any downside, incorporating it was a straightforward and highly beneficial choice. However, our core contribution remains the four points as illustrated in the Introduction.
> > > - **Method Design versus Extra Data:** To address the concern that the improvement might stem mostly from extra training data rather than the method itself, we highlight the comparison with other RL post-training methods in **Appendix C.6 and Table 9**. We evaluated existing proxy tasks, simply applying RL with extra data using these other methods, yields marginal gains and sometimes even negative returns on medical VQA benchmarks. However, our Med-Scout framework achieves significantly higher accuracy. This proves that merely adding training data is insufficient; the key driver of our performance boost lies entirely in our domain-specific geometric alignment design.
> > > - **Computational Load:** The calculation for our reward function relies solely on simple text extraction from the generated answers, followed by **basic mathematical arithmetic**. It completely avoids complex formulas. Consequently, the computational overhead is practically negligible compared to traditional GRPO. **Given that this approach yields consistent performance gains at virtually zero additional compute cost, we firmly believe its implementation is highly justified.**
> > > - **Future Manuscript Revisions:** We completely agree with your valuable suggestion to present both the standard GRPO baseline and our reward method in Table 1 for a clearer head-to-head comparison. Given the strict character limits and time constraints of this rebuttal phase, we commit to fully updating the tables and including expanded ablation studies on the reward functions in the final revised manuscript.
> > >
> > > We deeply appreciate your professionalism and the highly constructive feedback you have provided to improve our manuscript. We are eager to incorporate your valuable insights to further refine our work. Given the strict constraints of the rebuttal phase, we might not be able to exhaustively complete and present every additional experiment at this exact stage, and we sincerely hope for your understanding. Building upon our current responses, we remain fully committed to addressing your concerns and striving for a more positive evaluation. Thank you!

---

### Official Review · Reviewer_kAFN · 2026-03-12

**Soundness:** 2
**Presentation:** 2
**Significance:** 3
**Originality:** 3
**Overall Recommendation:** 3
**Confidence:** 4

**Summary:**

This paper studies a limitation of Multimodal Large Language Models for medical imaging, which the authors describe as insufficient geometric grounding. To motivate this problem, the paper presents a pilot study identifying three recurring failure modes: inconsistency across visual scales, incorrect reasoning about relative spatial relationships under transformations, and weak sensitivity to structural anomalies.

To address these issues, the paper introduces Med-Scout, a geometry-aware RL post-training framework for medical MLLMs. They convert unlabeled medical images into supervision through three proxy tasks: Hierarchical Scale Localization, Topological Jigsaw Reconstruction, and Anomaly Consistency Detection. The authors optimize models on these tasks using GRPO with a geometric reward designed to encourage stronger alignment between generated responses and the geometric structure of the image. The paper also introduces Med-Scout-Bench, a curated benchmark that consists of a balanced, high-quality subset of the constructed dataset.

Experiments compare the fine-tuned models with a range of open-source and proprietary baselines on Med-Scout-Bench, as well as on downstream medical VQA and report-generation tasks, to demonstrate that improving geometric perception can also benefit broader medical image understanding.

**Compliance With Llm Reviewing Policy:**

Affirmed.

**Final Justification:**

The rebuttal includes additional experiments that address some of my concerns. However, the pilot study remains unconvincing, and the gains on external benchmarks are small, which raises questions about the effectiveness of the proposed proxy tasks, especially when compared to the base GRPO.

**Key Questions For Authors:**

1. Could the authors clarify what percentage of evaluation failures are due to formatting issues versus actual perception errors for each evaluated model?
2. Could the authors clarify how the final subset used to construct Med-Scout-Bench was selected from the full generated dataset, and what criteria were used for this selection?
3. Could the authors clarify the evaluation settings used in the pilot study? For example, do they use CoT prompting?

**Limitations:**

yes

**Strengths And Weaknesses:**

**Strengths**

1. The paper tackles the timely and important issue of geometric blindness in medical MLLMs, with a well-motivated problem formulation.
2. The three proxy tasks for extracting geometric supervision from unlabeled medical data are creative and valuable.
3. The Dense Geometric Reward (DGR) within GRPO is a meaningful contribution that provides dense feedback for spatial reasoning.
4. The evaluation covers diverse benchmarks (VQA and report generation), strengthening empirical validation.
5. The generalization correlation analysis provides important empirical evidence that training on the proposed proxy tasks translate to gains on downstream medical benchmarks.

**Weakneses**
1. The pilot study lacks sufficient experimental detail (e.g., evaluation protocol, prompting strategy, and precise metric definitions), making it difficult to assess the central claim.
2. The baseline comparison may not be fully fair. It appears that base models are evaluated without Chain-of-Thought (CoT) reasoning, which is known to often improve performance. Allowing reasoning for the baseline models would provide a more convincing comparison.
3. Results on Med-Scout-Bench are mainly presented through a visually dense figure reporting aggregated reward scores. Clear tabular results with per-task metrics would improve interpretability.
4. The open-source baselines are limited to models up to 8B parameters. Evaluating larger models would help determine whether the observed geometric blindness persists at scale.
5. The claim in Finding 3 is not fully substantiated. Because the edited images differ from the natural image distribution used to train MLLMs, the observed performance drop may stem from out-of-distribution effects rather than a deficiency in geometric reasoning.
6. The reasoning reward mainly enforces output formatting rather than evaluating the logical correctness of the reasoning itself, which limits its ability to meaningfully supervise reasoning behavior.
7. The evaluation relies on open-ended answering, and it is unclear how many errors arise from formatting issues rather than actual failures in understanding.

---

> ### Author Rebuttal · Authors · 2026-03-31
>
> We sincerely thank Reviewer kAFN for the detailed review. We address the reviewer's comments and questions below:
>
> **1. Formatting Issues (KQ1 & W7)**
>
> **Response:**
>
> We believe formatting issues have a negligible impact on our evaluation, as supported by two key points:
>  - **LLM-Assisted Evaluation Pipeline:** Our standardized pipeline uses advanced LLMs (both for answer extraction and judgment) **with format checking instead of rigid string-matching (Please take Appendix A.4 for example)**. This ensures the evaluation focuses entirely on actual semantic and geometric correctness, naturally bypassing any minor formatting deviations.
>  - **Manual Audit Confirmation:** During this rebuttal period, we **manually audited** 500 randomly sampled cases per experiment (roughly a 5% subset). This audit revealed exactly a 0% failure rate caused by formatting errors.
>
> **2. Med-Scout-Bench Subset Selection Criteria (KQ2)**
>
> **Response:**
>
> The $10.8\text{K}$ Med-Scout-Bench subset was rigorously selected from the full $108\text{K}$ pool based on two core criteria to ensure an unbiased and comprehensive evaluation, as detailed in **Appendix A.1**.
>
> **3. Pilot Study Experimental Settings (KQ3 & W1)**
>
> **Response:**
>
> Due to space constraints, the detailed experimental settings adopted in the pilot study are presented in a figure via **[anonymous link](https://anonymous.4open.science/r/0129-00F2/)**, **we did not use CoT but used LLM-as-a-Judge**. We will include these details in the main text or the appendix in the final version.
>
> **4. Baseline Comparison (W2)**
>
> **Response:**
>
> We trained four CoT Med-Scout variants solely to verify textual reasoning on proxy tasks, as illustrated in **Section 5.5 and Figure 4**. However, across all benchmark evaluations, both our models and all baselines were tested strictly without CoT (e.g., disabling closed-source models' "thinking modes"). Thus, our comparison is entirely fair.
>
> **5. Detailed Benchmark Tabular Data (W3)**
>
> **Response:**
>
> Detailed tabular results are actually provided in **Table 5 of Appendix C.1**. We apologize if this was not sufficiently prominent in the main text due to space constraints.
>
> **6. Evaluation on Larger Models (W4)**
>
> **Response:**
>
> While our initial baselines focused on $<10$B models to demonstrate how our Med-Scout effectively mitigates geometric blindness at a smaller scale, we agree that this is insufficient for a comprehensive benchmark. To address this, we have evaluated a diverse set of **larger open-source models ($>10$B)** on Med-Scout-Bench to investigate whether this persists at scale. These new results are available via **[anonymous link](https://anonymous.4open.science/r/0129-00F2/)** and will be incorporated into the revised manuscript, and we will extend to more baselines in the future.
>
> **7. OOD Effects in Finding 3 vs. Geometric Reasoning (W5)**
>
> **Response:**
>
> We thank the reviewer for raising the concern about potential OOD effects. However, we believe the failure in Finding 3 shows a true lack of geometric reasoning, not just an OOD issue, for three main reasons:
> - **Natural Image Design:** As detailed in **Appendix A.2.3**, we carefully designed the edited images to look natural. We used real, highly similar medical patches (such as adjacent slices or top semantic retrievals) and blended the edges to avoid unnatural "cut-and-paste" artifacts.
> - **Improvements on Real Data:** If the models were simply failing because the images looked fake or unnatural, training on them wouldn't help the model understand normal images. Yet, as shown in our ablation study (**Appendix C.4, Table 7**), applying RL Alignment using _only_ this anomaly task ($\mathcal{T}_\text{anom}$) significantly improves performance on real, unedited medical VQA datasets. This proves our task fixes a genuine geometric weakness, rather than just teaching the model to spot artificial edits.
> - **Real-World Robustness:** We believe advanced MLLMs must generalize beyond pristine, lab-environment data distributions. Since artificially perturbed or corrupted images are realistic occurrences in real-world settings, a reliable model must possess robust visual perception to explicitly detect and report such anomalies, rather than failing silently when encountering imperfect data.
>
> **8. Limitation of the Reasoning Reward regarding Logical Correctness (W6)**
>
> **Response:**
>
> Thanks for this meaningful suggestion. **While explicitly rewarding the logical validity of the CoT process is an intuitively appealing strategy, our internal experiments actually yielded suboptimal results.** This underperformance suggests that enforcing explicit textual reasoning offers very limited benefits for strong visual tasks that fundamentally rely on rigorous, low-level geometric perception rather than semantic logic. Detailed metrics from these ablation experiments are available via **[anonymous link](https://anonymous.4open.science/r/0129-00F2/)** and will be incorporated into the revised appendix.

---

> > ### Author Rebuttal · Reviewer_kAFN · 2026-04-03
> >
> > Thank you for the rebuttal and for providing additional experiments and clarifications. I still have some remaining concerns.
> >
> > 3- While the additional details on the pilot study improve clarity, I still have concerns about the depth of the evaluation. The study relies on a single prompting strategy without CoT or robustness analysis, making it unclear whether the observed failures explicitly reflect true geometric limitations.
> >
> > 4- Although the evaluation is now clarified to exclude CoT for all models, I believe this may underestimate the capabilities of strong baselines. Including CoT-enabled evaluations would provide a more complete assessment of whether the observed limitations persist under stronger inference settings.
> >
> > 5- I acknowledge that detailed tabular results are included in the appendix. However, since it is an important table for the paper, I suggest the authors move it to the main paper for the final version, as it significantly improves the presentation.
> >
> >
> > 7- I appreciate the argument, but I am not fully convinced by the claim that if the images were unnatural, improvements on real data would not be observed after training. While I acknowledge the gains on external datasets, these improvements are relatively modest compared to those on Med-Scout-Bench. This suggests that part of the improvement may stem from adapting to the synthetic data distribution rather than purely improving geometric reasoning.
> >
> > 8- The discussion on rewarding logical correctness in CoT is convincing. However, I was unable to locate the corresponding experiments in the provided link. It would be helpful if the authors could explicitly point to the relevant section.

---

> > > ### Author Response · Authors · 2026-04-04
> > >
> > > We are deeply grateful for your ongoing feedback, which has greatly helped us refine our manuscript.
> > >
> > > **Question 3: Prompting Strategy without CoT**
> > >
> > > We agree that incorporating CoT prompting will provide a more rigorous assessment of the models' geometric reasoning capabilities, so we update the pilot study experiments:
> > > |**Model**|**Experiment 1**|**Experiment 2**|**Experiment 3**|
> > > |---|---|---|---|
> > > |*No-CoT*|---|---|---|
> > > |Qwen3-VL-8B-Instruct|69.50%|18.00%|9.00%|
> > > |Lingshu-7B|80.50%|20.50%|8.50%|
> > > |*CoT*|---|---|---|
> > > |Qwen3-VL-8B-Instruct|67.00%|20.50%|12.00%|
> > > |Lingshu-7B|77.50%|22.00%|7.00%|
> > >
> > > This strongly proves that the failure stems from a fundamental deficit in geometric reasoning capabilities, rather than a simple prompting artifact.
> > >
> > > **Question 4: CoT-Enabled Evaluation**
> > >
> > > We have carefully considered your valuable suggestion. We agree that adding CoT baselines would provide a stronger evaluation of the models' true capabilities. **However, we argue that simply applying a general CoT prompt (e.g., _"Please think step by step"_) across all baselines would actually result in an unfair comparison:**
> > > - **Model Variations:** Open-source models (e.g., Qwen) often have separate "Instruct" and "Thinking" versions, which behave very differently.
> > > - **Hidden Logic:** Closed-source models use complex, hidden reasoning mechanisms that cannot be directly or fairly compared with open-source models.
> > >
> > > Given the short rebuttal window, we want to avoid providing misleading results based on superficial CoT prompts. Instead, we consider to add a detailed evaluation to ensure a convincing assessment.
> > >
> > > **Question 5: Tabular Results**
> > >
> > > Absolutely. We will move these tabular results into the main text.
> > >
> > > **Question 7: OOD Issue**
> > >
> > > We appreciate the reviewer's careful scrutiny. The concern regarding potential overfitting to the synthetic distribution is highly relevant. However, comprehensive empirical evidence in our paper confirms that the improvements stem from true geometric reasoning rather than distributional adaptation:
> > > - **Contextualizing the "Modest" Gains:** Med-Scout-Bench evaluates _pure_ geometric perception, hence the massive gains (+40%). In contrast, established external benchmarks (e.g., SLAKE, VQA-RAD) are highly saturated, comprehensive evaluations containing many non-geometric queries. **Achieving a consistent 3-4% absolute gain on these global benchmarks _solely_ by curing geometric blindness is highly substantial**, not merely modest.
> > > - **The SFT vs. RL Evidence (Appendix C.7):** If the models were simply overfitting to synthetic artifacts, SFT would generalize to external datasets. As shown in **Tables 10 & 11**, SFT achieves high scores on our synthetic bench but yields _negative transfer_ on real external datasets. Only our Med-Scout forces the model to align with the underlying geometric manifold, proving the gains are derived from reasoning, not shortcut learning.
> > > - **Independent Verification on Real Images (Section 5.7):** Our Energy Landscape analysis (**Fig. 6 & 13**) provides definitive proof on 100% unperturbed, real images from MIMIC-CXR. By successfully differentiating spatially factual reports from hallucinated ones on strictly real scans, we empirically prove that the acquired geometric grounding operates independently of any synthetic visual distribution.
> > >
> > > **Question 8: Link Location**
> > >
> > > Due to rebuttal constraints, we rely on captions to explain each figure and table. Please refer to the table below, captioned "Performance Comparison among Med-Scout No-CoT (M), CoT (MR), and CoT-Logical (MRL)."
> > >
> > > We are deeply grateful for your rigorous and constructive feedback, which has been instrumental in refining our manuscript. We hope that our responses across these two rounds of rebuttal have fully addressed your concerns and might encourage a more positive evaluation of our work. Thank you very much for your valuable time!

---

### Decision · Program_Chairs · 2026-04-30

**Decision:**

Accept (regular)

**Comment:**

Med-Scout addresses a failure mode in medical MLLMs: the tendency to generate fluent but spatially incoherent responses. The paper proposes geometry-aware RL post-training using three proxy tasks (hierarchical scale localisation, topological reconstruction, anomaly consistency detection) with a Dense Geometric Reward in a GRPO framework, and introduces Med-Scout-Bench as an evaluation testbed.

Scores:
kAFN: weak reject, confidence 4
2kGv: weak accept, confidence 4
x71u: weak accept, confidence 4
cNAw: weak reject, confidence, 4

The scores were split, with two reviewers moving to weak accept after the rebuttal and two retaining partial concerns. Reviewer 2kGv’s concerns about prompting strategy were addressed, and cNAw’s concerns about novelty and the Baseline+GRPO comparison are legitimate but somewhat in tension with the paper’s framing: the authors credibly explain that GRPO is adopted as a customised optimisation trick rather than the core contribution, and the separate geometric proxy task design is the claimed novelty. The authors’ characterisation of cNAw’s review as reflecting domain bias seems overstated, but the core rebuttal arguments about how their definition of geometric blindness differs from general geometry-awareness are substantive and I find them broadly convincing.

The paper would benefit from a Baseline+GRPO comparison table and cleaner language around what constitutes Med-Scout-Bench (it is a curated benchmark rather than newly collected data). I also note that the 40% improvement claim deserves more careful contextualisation in the presentation. On balance, however, the problem addressed is clinically meaningful, the proxy task design is creative, and the generalisation results across medical VQA are encouraging.